# Multimodal vision-language models with guided cross-attention for crisis event understanding

## Abstract

Understanding crisis events from social media posts to support response and rescue efforts often requires robust multimodal reasoning over both visual and textual content. However, existing models often struggle to fully leverage the complementary nature of these modalities, particularly in noisy and information-sparse settings. In this work, we propose a novel multimodal framework, CapFuse-Net, that integrates pretrained vision-language models (VLMs) with a guided fusion strategy for improved crisis event classification. We first augment textual input with VLM-generated image-grounded captions, providing richer context for textual reasoning. A Cross-Feature Fusion Module (CFM) is then used to fuse the original and generated text using cross-attention, followed by a Guided Cross-Attention module that enables fine-grained interaction between visual and textual features. To further refine this fusion, we incorporate a Differential Attention mechanism that enhances salient feature selection while suppressing noise. Extensive experiments on three crisis classification benchmarks demonstrate that our method consistently outperforms unimodal and standard multimodal baselines. In addition, an ablation study demonstrates the importance of each proposed component, in particular, the synergy between VLM-based captioning and attention-guided fusion. Finally, we present results for qualitative interpretability through Grad-CAM visualizations and robustness across diverse crisis scenarios.

## 1 Introduction

The Internet and social networks have become sources of real-time information and news broadcasts. During crisis events such as wildfires, hurricanes, floods, and tsunamis, people actively share updates, images, and videos on social networks. This can create a vast pool of data that can aid in humanitarian response and decision-making. Extracting important information from this ongoing stream of data can help make quick decisions and use resources more effectively. However, not all social media posts contain relevant or actionable information. Hence, it becomes essential to filter out noninformative content and identify meaningful posts that support crisis management efforts. This challenge has led to more research on analyzing multimodal data with social media data containing text, images, links, and videos. However, this analysis requires advanced methods to combine and understand all these different types of information.

Multimodal machine learning has shown immense potential across various applications, including sentiment analysis, misinformation detection, mental health prediction, sarcasm detection in memes, etc. Lai et al. (2023); Shu (2022); Chancellor & De Choudhury (2020); Bandyopadhyay et al. (2023); Ngiam et al. (2011); Vielzeuf et al. (2018); Kiela et al. (2018); Abavisani et al. (2019). Despite its advantages, multimodal learning remains challenging due to complex interactions between different modalities and the difficulty of aligning heterogeneous data sources Baltrusaitis et al. (2018). Prior research has shown that training multimodal classification networks is often more difficult than their unimodal counterparts due to issues such as modality imbalance, misalignment, and varying levels of noise in data

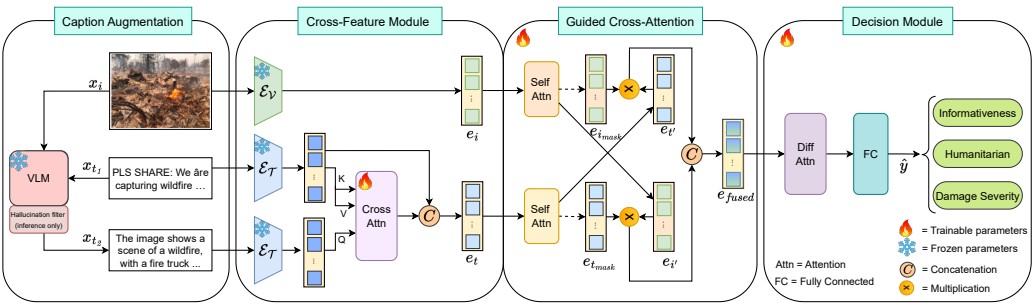

Figure 1: Proposed CapFuse-Net: the pipeline starts with Caption Augmentation using a VLM, where a hallucination filter (inference only) mitigates spurious information. VLM-generated caption and original text information are fused in the Cross-Feature Module to get enriched text feature; image and text features are aligned and fused in the Guided Cross-Attention; and refined in the Decision Module with Differential Attention for predictions on informativeness, humanitarian categories, and damage severity.

Wang et al. (2020). Addressing these challenges is crucial for building robust models capable of effectively supporting crisis response efforts.

Vision-language models (VLMs) have made significant progress in aligning image and text modalities across various tasks, including image captioning, visual question answering, and cross-modal retrieval Li et al. (2022); Alayrac et al. (2022); Yu et al. (2022); Liu et al. (2023). By learning to combine visual and text information, these models make it easier to integrate and understand multimodal data together, helping to improve interpretability and context Radford et al. (2021); Li et al. (2022); Liu et al. (2023). Models such as CLIP Radford et al. (2021), BLIP Li et al. (2022), Perception Encoder (PE) Bolya et al. (2025), and LlaVA Liu et al. (2023) leverage large-scale pretraining on diverse datasets to learn fine-grained visual and semantic associations from text and images and provide strong contextual representations. These models can be applied in crisis situations, where images often have complex information that can be difficult to understand without additional context. While CLIP and PE learn contrastive embeddings, generative VLMs such as BLIP and LLaVA can produce rich, free-form captions or dialogue about images, which can be leveraged to summarize complex crisis scenes into concise, actionable descriptions. We incorporate LLaVA to generate detailed image captions, enriching the textual information available for multimodal learning. This augmentation can provide more contextually relevant descriptions, therefore helping to reduce ambiguity in image-text alignment. We integrate this augmented caption into our classification pipeline using a Cross-Feature Fusion Module (CFM), inspired by the Cross-Adapter Module (CAM) Liu et al. (2025). In CAM, two pretrained models are employed to extract enriched feature representations from textual inputs, which are subsequently fused via cross-attention. In our proposed CFM, we similarly leverage pretrained models; however, we obtain features from both the original and augmented textual inputs and fuse them using a cross-attention mechanism. This enables the model to capture complementary semantic information from both views of the input, thereby enhancing the quality of the joint representation.

Furthermore, to advance multimodal learning for the classification of crisis events, we combine our proposed CFM architecture with the Guided Cross-Attention mechanism to refine the fusion process between the textual and visual modalities Gupta et al. (2024). We also employ a differential attention mechanism Ye et al. (2024) to emphasize more informational modalities while downweighting less relevant ones. Unlike conventional attention mechanisms Vaswani et al. (2017), which treat all modalities uniformly, our approach assigns adaptive attention weights based on the contextual importance of each modality in a given instance. This ensures that the most informative signals receive higher focus, leading to improved interpretability and classification accuracy. Our model effectively captures the intricate dependencies between images and textual descriptions, resulting in more precise crisis assessment and response strategies. Our contributions are summarized below.

- A novel multimodal reasoning model (CapFuse-Net) to enhance text-image alignment through caption augmentation and knowledge fusion

- An innovative fusion module (CFM) leveraging original and augmented textual inputs with cross-attention to enhance joint representation learning

- Extensive evaluation across multiple tasks on multimodal crisis and non-crisis benchmark datasets, demonstrating the superiority of our proposed CapFuse-Net over existing multimodal fusion methods

- Detailed analyses including comprehensive ablation studies showcasing the robustness, generalizability, and interpretability of the proposed model in understanding crisis events

## 2 Related Work

### 2.1 Multimodal Learning

Multimodal learning, which involves fusing information from multiple modalities such as text, images, audio, and video, has become increasingly prominent due to its ability to capture richer contextual information than unimodal approaches. Early fusion approaches concatenate raw or low-level features, while late fusion approaches combine predictions from separate unimodal models. However, these approaches often struggled to capture intricate inter-modal relationships Baltrusaitis et al. (2018). More recent advances, including attention mechanisms, such as the Transformer architecture (Vaswani et al., 2017), have been adapted to process and integrate multimodal data effectively Li et al. (2019); Huang et al. (2020); Kim et al. (2021); Kaduri et al. (2025). For example, ViLT (Vision and Language Transformer) (Kim et al., 2021), which processes image patches and textual tokens through a unified transformer without relying on a separate visual encoder, shows strong performance in tasks such as image-text matching and visual question answering. Similarly, UNITER (Chen et al., 2020) learns joint visual-linguistic representations using cross-modal attention and has achieved state-of-the-art results on several benchmark datasets. Parallel to this, contrastive learning has emerged as a powerful paradigm for learning aligned multimodal representations. CLIP Radford et al. (2021), ALIGN Jia et al. (2021), and Perception Encoder Bolya et al. (2025) align images and text in a shared embedding space using large-scale noisy web data and contrastive loss and demonstrate impressive zero-shot transfer across diverse vision-language tasks. Generative vision-language models have further pushed the field forward. BLIP Li et al. (2022), Flamingo Alayrac et al. (2022), CoCa (Yu et al., 2022), and LLaVA Liu et al. (2023) unify vision and language for generation tasks using encoder-decoder or instruction-tuned models. These models excel at captioning, visual question answering, and multimodal dialogue, especially when fine-tuned with instruction-following datasets.

### 2.2 Crisis Event Analysis

Crisis event analysis has emerged as a critical area of research in both computer vision and natural language processing, motivated by the need for timely and reliable situational awareness during natural disasters, conflicts, and humanitarian emergencies. The availability of diverse data sources such as satellite imagery, social media posts, and sensor feeds has driven the development of multimodal methods for detecting, classifying, and assessing crisis events. Early research in crisis informatics leveraged textual data from platforms such as Twitter to extract crisis-related information, including situational updates, resource needs, and damage reports (Olteanu et al., 2014). Crowd-sourced labeling and supervised classification techniques have proven to be effective for real-time filtering of relevant crisis tweets Imran et al. (2014). To improve information extraction and situational awareness, researchers have collected and shared multimodal datasets that combine textual and visual information. CrisisMMD (Alam et al., 2018; Ofli et al., 2020a) is one of the most comprehensive such resources, containing tweets paired with images, annotated for informativeness, humanitarian categories (e.g., infrastructure damage, injured people), and severity levels. This dataset facilitated the development of multimodal models that can align textual and

visual signals to improve classification and retrieval performance. Multimodal deep learning approaches have exploited both images and text to enhance crisis information processing. Early models often used convolutional neural networks (CNNs) for visual features and recurrent neural networks (RNNs) or transformers for textual data, with late or intermediate fusion strategies (Ofli et al., 2020b). An event detection model that integrated both low and high-level features to leverage their complementary strengths was proposed in prior work (Wang et al., 2021a). The study in Abavisani et al. (2020) highlights the challenges of aligning heterogeneous modalities and demonstrates that multimodal models outperform unimodal approaches in classifying disaster-related data. Building upon this, CrisisKAN Gupta et al. (2024) was proposed as a knowledge-augmented attention mechanism that enhances interpretability and classification accuracy. By incorporating external crisis-related knowledge from Wikipedia, CrisisKAN improved contextual understanding and robustness, particularly in low-resource scenarios Gupta et al. (2024). Complementing knowledge-based attention, the Multimodal Channel Attention (MMCA) mechanism was introduced to emphasize informative channels in textual and visual representations, thereby improving fusion effectiveness and achieving stronger crisis categorization performance Rezk et al. (2023). The above studies have demonstrated the importance of multimodal fusion and the integration of external knowledge in crisis response applications. More recent work, such as CaMN, introduced a cross-aligned multimodal framework that generates Abstract Meaning Representation (AMR) graphs and integrates visual, textual, and AMR graph representations using a cross-alignment loss and masked autoencoding Rajora et al. (2025). Another work proposed a multimodal extractive abstractive summarization model for crisis-related microblogs, combining pretrained vision-language models to fuse text and image information, achieving significant improvements Kumar et al. (2024).

## 3 METHOD

Our proposed multimodal architecture is shown in Fig. 1. For a multi-modal data instance $(x_i, x_t, y)$, where $x_i$ represents an image, $x_t$ represents tweet text, and $y$ is the task-specific class label. The unimodal models are designed to evaluate the individual contributions of the visual and textual modalities in the absence of multimodal fusion. The objective is to improve classification performance by leveraging VLMs for knowledge fusion and cross-modal alignment.

### 3.1 UNIMODAL MODEL

In the image-only model, we input an image $x_i$ into a frozen vision encoder $\mathcal{E}_V$, such as DenseNet Huang et al. (2017) or CLIP Vision Radford et al. (2021). The encoder produces a visual feature representation $e_i$, which is subsequently passed through a self-attention module to enhance relevant spatial and semantic features. The attended features are then forwarded to a multilayer perceptron (MLP) classifier to generate the predicted label $\hat{y}$. The text-only model follows a parallel structure but operates on textual input $x_t$, such as a tweet associated with the image. The text is encoded using a frozen text encoder $\mathcal{E}_T$. For robust analysis, we experiment with Electra Clark et al. (2020) and the CLIP Text model. The resulting text embedding $e_t$ is refined via a self-attention module and passed to an MLP classifier to produce the prediction $\hat{y}$. The unimodal model architecture is shown in the Appendix Fig. 4.

### 3.2 MULTI-MODAL MODEL

We employ a vision–language model (VLM) to generate descriptive text specific to images. A major concern in VLM-based augmentation is the tendency of these models to introduce hallucinations, where objects or attributes not present in the image are falsely described. Several approaches have been proposed to mitigate this problem Jiang et al. (2025); Liu et al. (2024). In our work, we adopt the method of Jiang et al. (2025), which leverages middle-layer attention consistency to detect and suppress hallucinations during the inference stage of LLaVA (implementation details are provided in the appendix E.1).

Following Jiang et al. (2025), we construct an instruction $I$ that incorporates the original text $x_{t_1}$. Details on the instruction $I$ can be found in the Appendix Section E. This instruction, along with the corresponding image $x_i$, is provided as input to the VLM (LLaVA). The VLM model then generates a descriptive caption $(x_{t_2})$ that reflects the visual content of the image. The caption generation process can be formally expressed as: $x_{t_2} = \text{VLM}(I, x_i)$.

CapFuse-Net is fully modular with respect to the captioning component. The use of LLaVA reflects a reproducible and widely available open-source choice rather than a strict dependency of the framework. The captioning step is performed once during offline pre-processing, so it does not impact the inference-time efficiency of the model. This modular design also enables the caption generator to be replaced with faster or higher-fidelity alternatives such as GPT-4o Achiam et al. (2023), Gemini Team et al. (2023) accessed through APIs, as well as other lightweight VLMs, which can offer improved latency, reduced hallucination rates, or more detailed descriptions depending on deployment requirements.

We also augment knowledge from Wikipedia following the CrisisKAN method Gupta et al. (2024). At this stage, we have two textual inputs; we denote the original text as $x_{t_1}$ and the VLM-generated image description as $x_{t_2}$. We employ the cross-feature module to effectively combine semantic information from both sources. We utilize the text encoder from a pretrained foundational multimodal $\mathcal{E}_{\mathcal{T}}$ (e.g., CLIP or PE) to extract text embeddings from both inputs. Specifically, the encoder generates enriched embeddings $e_{t_1}$ for the original text and $e_{t_2}$ for the generated caption, as defined below: $e_{t_1} = \mathcal{E}_{\mathcal{T}}(x_{t_1}), \quad e_{t_2} = \mathcal{E}_{\mathcal{T}}(x_{t_2})$.

In the cross-feature fusion module, we improve the interaction between the original text $x_{t_1}$ and the VLM-generated image caption $x_{t_2}$ by applying a cross-attention mechanism between their corresponding embeddings. We compute the cross-attentive feature representation $e_{t'}$ by using the embedding of the VLM-generated caption $e_{t_2}$ as the query, while the embedding of the original text $e_{t_1}$ serves as both the key and value in the cross-attention mechanism:

$$e_{t'} = \text{Cross-Attn}(Q = e_{t_2}, \ K = e_{t_1}, \ V = e_{t_1}). \tag{1}$$

To obtain the final fused textual representation $e_{t'}$, we concatenate the attended feature $e_{t_2}$ with the original text embedding $e_{t_1}$. We introduce a scaling factor $\lambda$ to control the contribution of the cross-attentive feature. Here, we set $\lambda = 0.5$, which serves as a balancing parameter that amplifies the informative signals from cross-fusion while suppressing less relevant information, thereby ensuring a more robust textual representation. $e_t = \text{Concat}(\lambda \cdot e'_t, \ e_{t_1})$.

Here, $\lambda$ is a hyperparameter that balances the influence of VLM-generated information relative to the original text. In addition to CFM, we also experiment with a simple concatenation of the original text and the VLM-generated text and pass the concatenated text to the text encoder to obtain the combined textual representation $e_t$. In parallel, a vision encoder $\mathcal{E}_{\mathcal{V}}$ from a pre-trained foundation model is used to extract the visual feature representation $e_i$ from the input image $x_i$: $e_i = \mathcal{E}_{\mathcal{V}}(x_i)$.

We freeze all pre-trained models during training to maintain the conceptual alignment between the text and image modalities. Next, we apply the Guided Cross-Attention module, following the approach proposed in Gupta et al. (2024). We begin by applying self-attention independently to each modality (image and text). Given a feature vector $V$ of $d$ dimention, we compute self-attention as:

$$\text{Self-Attn}(V) = \text{softmax}\left(\frac{VV^T}{\sqrt{d}}\right) V. \tag{2}$$

This operation enhances contextual understanding within each modality before cross-modal fusion. After applying self-attention to the image and text feature vectors $e_i$ and $e_t$, we obtain their updated representations:

$$z_i = \text{Self-Attn}(e_i), \quad z_t = \text{Self-Attn}(e_t). \tag{3}$$

Next, we compute new projections of the image and text features along with their corresponding attention masks. For each modality, we apply a linear transformation followed by an activation function $F$, and compute the attention mask using the sigmoid function $\sigma$:

$$e'_i = F(W_i z_i + b_i), \quad e_{i_{mask}} = \sigma(W'_i z_i + b_i). \tag{4}$$

We follow a similar process for the text modality:

$$e'_t = F(W_t z_t + b_t), \quad e_{t_{mask}} = \sigma(W'_t z_t + b_t). \tag{5}$$

Next, we apply Guided Cross-Attention by modulating each modality's projected feature with the attention mask of the other modality. Specifically, we multiply the attention mask of the text modality $e_{t_{mask}}$ with the projected image feature $e'_i$, and the attention mask of the image modality $e_{i_{mask}}$ with the projected text feature $e'_t$. This cross-modulation enriches the feature representations by incorporating complementary information from the other modality. Finally, we concatenate the resulting attention-weighted features to obtain the joint multimodal representation: $e_{fused} = \text{concat}(e_{t_{mask}} \cdot e'_i, \; e_{i_{mask}} \cdot e'_t)$.

After obtaining the final fused representation by concatenation, we apply a differential attention mechanism to further refine the feature vector Ye et al. (2024). Given the input matrix $X \in \mathbb{R}^{N \times d_{\text{model}}}$, we compute the query, key, and value projections using learnable parameter matrices:

$$[Q_1; \; Q_2] = XW^Q, \quad [K_1; \; K_2] = XW^K, \quad V = XW^V, \tag{6}$$

Here, $W^Q, W^K, W^V \in \mathbb{R}^{d_{\text{model}} \times 2d}$ are the weight matrices for the query, key, and value projections, respectively. We then compute the differential attention output by subtracting a scaled secondary attention from the primary attention, as follows:

$$\text{DiffAttn}(X) = \left( \text{softmax}\left( \frac{Q_1 K_1^T}{\sqrt{d}} \right) - \gamma \cdot \text{softmax}\left( \frac{Q_2 K_2^T}{\sqrt{d}} \right) \right) V, \tag{7}$$

Here, $\gamma$ is a learnable scalar parameter that controls the strength of the differential signal. This mechanism enables the model to capture contrastive patterns between different attention pathways, promoting more discriminative feature learning.

We then obtain the final refined multimodal representation $z$ by applying the differential attention mechanism to the fused feature vector $z = \text{DiffAttn}(e_{fused})$. Finally, we pass this refined representation through a fully connected classification head, followed by a softmax layer, to produce the predicted class label $\hat{y} = \text{softmax}(FC(z))$.

## 4 EXPERIMENTAL EVALUATION

### 4.1 DATA

We utilize three different datasets to evaluate our proposed CapFuse-Net model. **CrisisMMD:** We evaluate our model on three different tasks by using image-text pairs of crisis events from the CrisisMMD dataset Alam et al. (2018). The dataset includes manual annotations for Task 1: binary classification of informative vs. non-informative posts, Task 2: categorization into five humanitarian classes (infrastructure and utility damage, vehicle damage, rescue volunteering or donation efforts, affected individuals, other relevant information), and Task 3: assessment of damage severity (severe, mild, or little/no damage). We follow the evaluation framework of Abavisani et al. (2020) and adopt the train/validation/test splits from Gupta et al. (2024). **DMD:** To test the generalizability of the model, we use 4,882 image–text pairs from the Damage Multimodal Dataset (DMD) dataset Mouzannar et al. (2018). Due to Task 2 label inconsistency with CrisisMMD, we focus on Tasks 1 and 3 for evaluation. **N24News:** We validate CapFuse-Net using 6,125 samples (train:

Table 1: Quantitative comparison against state-of-the-art multimodal models on the Crisis-MMD tasks. Best results are **Bolded** and second best results are underlined.

| Model | Task 1 | | | Task 2 | | | Task 3 | | |
|---|---|---|---|---|---|---|---|---|---|
| | Accuracy | Macro F1 | W-F1 | Accuracy | Macro F1 | W-F1 | Accuracy | Macro F1 | W-F1 |
| MMBT Kiela et al. (2019) | 86.4 | 85.3 | 86.2 | 88.7 | 64.9 | 89.6 | 70.1 | 59.2 | 68.7 |
| GMU Arevalo et al. (2017) | 87.2 | 84.6 | 85.7 | 88.7 | 64.3 | 89.1 | 70.6 | 57.1 | 68.2 |
| ViLT Kim et al. (2021) | 87.6 | 85.1 | 88.0 | 86.7 | 61.2 | 87.2 | 67.6 | 58.4 | 65.0 |
| CentralNet Vielzeuf et al. (2018) | 87.8 | 85.3 | 86.1 | 89.3 | 64.7 | 89.8 | 71.1 | 57.4 | 68.7 |
| CBGP Kiela et al. (2018) | 88.1 | 86.7 | 87.3 | 84.7 | 65.1 | 88.7 | 67.9 | 50.7 | 64.6 |
| VisualBERT Li et al. (2019) | 88.1 | 86.7 | 88.6 | 87.5 | 64.7 | 86.1 | 66.3 | 56.7 | 62.1 |
| ViLBERT Lu et al. (2020) | 88.4 | 86.5 | 88.7 | 88.2 | 65.1 | 86.6 | 65.9 | 56.3 | 61.8 |
| TinyCLIP Wu et al. (2023) | 84.2 | 81.1 | 83.7 | 86.7 | 59.5 | 86.4 | 64.8 | 41.2 | 56.0 |
| PixelBERT Huang et al. (2020) | 88.7 | 86.4 | 87.1 | 89.1 | 66.5 | 88.9 | 65.2 | 57.3 | 63.7 |
| Cross-attention Abavisani et al. (2020) | 88.4 | 87.6 | 88.7 | 90.0 | 67.8 | 90.2 | 72.9 | 60.1 | 69.7 |
| MCAModel Rezk et al. (2023) | 89.0 | 87.2 | 88.9 | 89.7 | 59.3 | 89.7 | 62.3 | 43.5 | 58.4 |
| UniS-MMC Zou et al. (2023) | 90.9 | 89.6 | 90.2 | 88.7 | 68.1 | 88.6 | 70.7 | 58.1 | 69.5 |
| CrisisKAN Gupta et al. (2024) | 86.8 | 85.3 | 86.9 | 91.3 | 66.1 | 91.2 | 64.7 | 44.6 | 61.0 |
| CaMN Rajora et al. (2025) | 92.8 | 91.3 | 92.7 | 92.4 | 67.5 | 92.2 | **73.2** | 60.7 | **71.1** |
| **CapFuse-Net (Ours)** | **93.6** | **92.8** | **93.6** | **95.7** | **71.5** | **95.3** | 71.1 | **60.9** | 69.5 |

Table 2: Comparison of unimodal and multimodal models on the CrisisMMD dataset using different visual encoders, textual encoders, knowledge fusions (KF), and cross-modal fusion strategies. [CA = Cross-Attention, Diff Attn = Differential Attention, CFM = Cross-Feature Fusion Module]. Detailed results in Table 8.)

| Modality | Vision | Text | KF | Fusion | Accuracy | | |
|---|---|---|---|---|---|---|---|
| | | | | | Task 1 | Task 2 | Task 3 |
| Image-only | DenseNet | - | - | - | 82.89 | 86.25 | 62.57 |
| Image only | CLIP Vision | - | - | - | 89.20 | 91.43 | 69.50 |
| Image only | PE Vision | - | - | - | 91.07 | 91.28 | 70.70 |
| Text-only | - | Electra | Wiki | - | 84.64 | 87.36 | 62.45 |
| Text only | - | CLIP text | tweet | - | 87.33 | 86.10 | 61.81 |
| Text only | - | CLIP text | Wiki | - | 83.53 | 83.15 | 59.48 |
| Text only | - | CLIP text | LLaVA | - | 87.64 | 86.33 | 62.13 |
| Multi-modal | DenseNet | Electra | Wiki | Guided CA | 86.80 | 91.34 | 64.65 |
| Multi-modal | DenseNet | Electra | Wiki | Cross Attention | 87.32 | 89.28 | 63.07 |
| Multi-modal | DenseNet | Electra | Wiki | Guided CA+Self Attn | 88.36 | 91.43 | 63.83 |
| Multi-modal | DenseNet | Electra | Wiki | Cross Diff Attn | 85.74 | 86.55 | 61.69 |
| Multi-modal | DenseNet | Electra | Wiki | Guided CA+Diff Attn | 89.33 | 91.58 | 63.14 |
| Multi-modal | CLIP Vision | CLIP text | Wiki | Guided CA | 90.57 | 94.02 | 68.94 |
| Multi-modal | CLIP Vision | CLIP text | Wiki | Guided CA+Diff Attn | 90.44 | 93.72 | 68.68 |
| Multi-modal | CLIP Vision | CLIP text | LLaVA | Guided CA+Diff Attn | 92.52 | 93.87 | 68.87 |
| Multi-modal | CLIP Vision | CLIP text | LLaVA | Guided CA | 92.91 | 93.92 | 69.00 |
| Multi-modal | PE Vision | PE text | LLaVA | Guided CA | 91.29 | 94.38 | 70.89 |
| Multi-modal | PE Vision | CLIP text | LLaVA | Guided CA | 92.98 | 94.97 | 70.19 |
| Multi-modal | CLIP Vision | CLIP text | LLaVA | CFM + Guided CA | 92.33 | 94.01 | 70.51 |
| Multi-modal | PE Vision | CLIP text | LLaVA | CFM + Guided CA | **93.63** | **95.72** | **71.14** |

4,899, validation: 613, test: 613) from the multimodal news article dataset Wang et al. (2021b). Please refer to Section D for the dataset details.

## 4.2 EXPERIMENTAL SETUP & EVALUATION METRICS

We trained our model on all three tasks using the SGD optimizer with categorical cross-entropy loss for 50 epochs. All experiments were carried out with a base learning rate of $1 \times 10^{-3}$ and a batch size of 16. To reduce overfitting, we applied early stopping based on validation loss, with a patience of five epochs. We implemented all models in Python using the PyTorch framework and ran our experiments on a machine equipped with an Intel(R) Xeon(R) processor, 128 GB of RAM, and two NVIDIA RTX A4000 GPUs. For model training, we trained the entire network when using DenseNet and ELECTRA as encoders, whereas we kept the pre-trained encoders frozen when using CLIP or PE. We evaluated performance using accuracy, macro F1 score, and weighted F1 score, and we averaged the results over three independent runs.

## 4.3 RESULTS AND DISCUSSION

Table 1 presents a comprehensive evaluation of various state-of-the-art multimodal models across three crisis classification tasks, comparing their performance in terms of accuracy, macro F1, and weighted F1 scores. Our proposed CapFuse-Net model, CFM with LLaVA-augmented caption, demonstrates consistent and superior performance in Tasks 1 and 2,

Table 3: Ablation study on CrisisMMD, evaluating the impact of text input from LLaVA and LLaVA-mitigated (with hallucination mitigation applied), the Cross-Feature Fusion Module (CFM), and Differential Attention (Diff Attn), with results reported in accuracy.

| Text | CFM | Diff Attn | Task 1 | Task 2 | Task 3 |
|---|---|---|---|---|---|
| LLaVA | ✓ | ✓ | $93.11 \pm 0.27$ | $94.95 \pm 0.53$ | $70.25 \pm 1.27$ |
| LLaVA | ✓ | ✗ | $\mathbf{93.63 \pm 0.29}$ | $\mathbf{95.72 \pm 0.34}$ | $71.14 \pm 0.93$ |
| LLaVA-mitigated | ✓ | ✗ | $93.37 \pm 0.21$ | $95.57 \pm 0.38$ | $\mathbf{71.96 \pm 1.43}$ |
| LLaVA-mitigated | ✓ | ✓ | $93.02 \pm 0.20$ | $94.16 \pm 0.13$ | $71.39 \pm 0.66$ |

achieving the highest scores across all three evaluation metrics. In Task 1, CapFuse-Net achieves an accuracy of 93.6%, macro F1 of 92.8%, and weighted F1 of 93.6%, outperforming all other baselines, including recent models such as UniS-MMC Zou et al. (2023) and CaMN Rajora et al. (2025). This result shows the robustness and generalization of the model across class distributions. In Task 2, which is particularly challenging due to the lower macro F1 values observed across models, CapFuse-Net again leads with an accuracy of 95.7%, macro F1 of 71.5%, and weighted F1 of 95.3%. Notably, the macro F1 improvement indicates the model's effectiveness in addressing class imbalance and performing well across all categories, not just the dominant ones. In Task 3, while CaMN slightly outperforms our model in terms of accuracy and weighted F1, the CapFuse-Net achieves the highest macro F1, suggesting that it provides a better balance across all classes. Given that macro F1 is particularly sensitive to the performance on minority classes, this result confirms the ability of our approach to provide equitable performance across different classes. To summarize, these results demonstrate that our proposed CapFuse-Net significantly enhances the model's ability to learn meaningful multimodal representations across varied crisis classification scenarios, particularly in tasks with imbalanced class distributions.

The results in Table 2 comparing unimodal and multi-modal model results show that image-based models outperform text-only approaches. CLIP leverages aligned visual-textual embeddings to capture semantically meaningful patterns in disaster imagery. Meanwhile, PE Vision achieves the best image-only performance, even surpassing some multimodal baselines, underscoring its strong visual representations. In contrast, text-only models (e.g., Electra, raw tweets) underperform due to the brevity and ambiguity of tweets. Yet, VLM-generated captions substantially boost results. For instance, CLIP Text with LLaVA captions yields better results over Wiki inputs, showing the benefit of visually grounded language. The best model, CapFuse-Net, integrates the full fusion stack: CFM, Guided Cross-Attention, and VLM-generated text, enabling joint reasoning over explicit tweet content and implicit visual semantics. This layered fusion delivers significant gains over the strongest unimodal baselines, producing more balanced and robust humanitarian classification. The differential attention was included to explore deeper refinement of subtle crisis cues. Results in Table 2 show that while it helps in some settings, it does not consistently yield improvements and is therefore omitted from the final model configuration.

## 4.4 ABLATION STUDY

**Module-wise analysis of CapFuse-Net** Table 3 shows the impact of text input choice and fusion modules in CapFuse-Net. Using only raw tweets yields the lowest performance, reflecting the limitations of short and noisy text. Adding LLaVA captions improves results, particularly for Task 3, demonstrating the value of visually grounded text. With LLaVA input, CFM proves essential: the best overall balance is achieved when CFM is retained without Differential Attention, reaching 95.72% on Task 2 and 71.14% on Task 3. Further applying hallucination mitigation (explained in the Appendix E.1) during caption generation slightly raises Task 3 accuracy to 71.96%, but the gap with unmitigated LLaVA is small, indicating that CFM effectively gathers salient information while discarding noisy signals. Overall, the comparison between LLaVA and LLaVA-mitigated captions suggests that hallucination suppression produces only a small numerical change in Task 3 and virtually no effect on Tasks 1 and 2. This indicates that the hallucinated elements present in the original captions of the datasets do not materially alter the predictions.

Table 4: Robustness analysis under different data sampling demonstrates the superiority of our CapFuse-Net for both CrisisMMD and DMD datasets. We report accuracy scores and percentage performance loss (in subscript)/gain (in superscript) from the original split.

| Split | Model | CrisisMMD Dataset | | | DMD Dataset | |
|---|---|---|---|---|---|---|
| | | Task 1 | Task 2 | Task 3 | Task 1 | Task 3 |
| Original | CapFuse-Net | 93.63 | 95.72 | 71.14 | 89.78 | 45.26 |
| | CrisisKAN | 86.80 | 91.34 | 64.65 | 81.35 | 23.62 |
| Random | CapFuse-Net | $92.83_{0.85\%}$ | $93.70_{2.11\%}$ | $69.87_{1.78\%}$ | $91.93^{2.39\%}$ | $41.53_{8.24\%}$ |
| | CrisisKAN | $85.40_{1.61\%}$ | $86.30_{5.52\%}$ | $63.82_{1.28\%}$ | $82.69^{1.64\%}$ | $22.86_{3.22\%}$ |
| Stratified | CapFuse-Net | $92.66_{1.04\%}$ | $94.29_{1.49\%}$ | $70.44_{0.98\%}$ | $88.18_{1.78\%}$ | $50.77^{12.17\%}$ |
| | CrisisKAN | $84.87_{2.22\%}$ | $85.89_{5.97\%}$ | $64.31_{0.53\%}$ | $84.38^{3.72\%}$ | $21.34_{9.65\%}$ |
| Event-wise | CapFuse-Net | $89.70_{4.19\%}$ | $94.04_{1.75\%}$ | $66.22_{6.91\%}$ | $86.88_{3.23\%}$ | $47.58^{5.13\%}$ |
| | CrisisKAN | $80.32_{7.47\%}$ | $84.66_{7.31\%}$ | $60.93_{5.75\%}$ | $83.25_{2.34\%}$ | $31.84^{34.80\%}$ |

Table 5: Evaluation on the N24News dataset suggesting the superiority of CapFuse-Net over the compared methods even in non-crisis tasks.

| Model | Accuracy | Macro-F1 | W-F1 |
|---|---|---|---|
| Cross-Attention Abavisani et al. (2020) | 43.55 | 38.84 | 43.04 |
| CrisisKAN Gupta et al. (2024) | 29.03 | 24.05 | 26.53 |
| MCAModel Rezk et al. (2023) | 60.00 | 52.07 | 58.14 |
| CapFuse-Net | **69.00** | **66.47** | **68.28** |

**Generalization across dataset and splits.** We evaluated model robustness across different splits of CrisisMMD and further tested transferability to an out-of-distribution DMD dataset, which lies in the same crisis domain but presents different distributions. As shown in Table 4, the CapFuse-Net consistently outperformed CrisisKAN on CrisisMMD, achieving 93.63% on Task 1, 95.72% on Task 2, and 71.14% on Task 3 in the original split. These gains of 6–7% indicate that CapFuse-Net captures richer multimodal representations, leading to stronger performance across all tasks. Under random and stratified splits, both models experienced mild accuracy losses relative to the original split, but CapFuse-Net's degradation was consistently smaller than CrisisKAN's. This shows that CapFuse-Net is more stable under distributional shifts. The event-wise split posed the greatest challenge since models must generalize to entirely unseen crisis events rather than overlapping scenarios. Stronger performance in this setting, where CapFuse-Net exceeded CrisisKAN by 9.4% on Task 1 and 5.3% on Task 3, shows its ability to transfer knowledge to novel emergencies, which is critical for real-world crisis response. For the cross-dataset evaluation, we trained on CrisisMMD (under each split) and tested on DMD. Here, CapFuse-Net not only preserved higher absolute accuracy than CrisisKAN but also showed smaller performance drops under resampling. These patterns highlight that although distribution shifts inevitably cause some losses, CapFuse-Net not only achieves higher baseline accuracy but also adapts more stability across splits, resulting in stronger transferability to unseen datasets like DMD.

**Generalization study on CapFuse-Net.** Table 5 presents the results in terms of Accuracy, Macro-F1, and Weighted F1. Cross-attention achieves moderate performance, while CrisisKAN struggles to adapt to this domain due to its crisis-specific design. MCAModel provides a stronger baseline with 60% Accuracy and 52.07 Macro-F1. In contrast, CapFuse-Net achieves 69% Accuracy, 66.47 Macro-F1, and 68.28 Weighted F1, outperforming the strongest baseline by over 14% in Macro-F1. The superior performance of CapFuse-Net can be attributed to its Cross-Feature Fusion Module (CFM), which captures complementary information between image and text representations more effectively than simple cross-attention. Additionally, the guided attention mechanism mitigates irrelevant or noisy signals from either modality, which is particularly beneficial for news articles where images may only weakly correlate with textual content. These design choices enable CapFuse-Net to generalize beyond crisis data and perform strongly in fine-grained multimodal classification.

Table 6: Comparison of trainable and total parameters for different models.

| Model | Trainable Parameters | Total Parameters |
|---|---|---|
| MCAModel (Task 1) | 165,401,607 | 165,401,607 |
| CrisisKAN (Task 1) | 129,301,179 | 129,301,179 |
| CapFuse-Net (Task 1) | 2,958,339 | 513,783,812 |

**Computational Efficiency.** As reported in Table 6, CapFuse-Net requires only 2.96M trainable parameters, which is substantially smaller than the baseline CrisisKAN (129M) and MCAModel (165M) models. Despite integrating multiple fusion components, the model maintains efficient inference, operating at approximately 0.12 to 0.13 seconds per minibatch of 16 samples across all three tasks. This low latency reflects the lightweight nature of the fusion layers and the fact that the backbone encoders remain frozen during training. Importantly, the LLaVA captioning step in our experiments is performed once during offline preprocessing, which removes this cost from the reported inference times. In real-time deployments where captioning must occur on the fly, CapFuse-Net remains compatible with faster captioning backends such as GPT-4o Achiam et al. (2023), Gemini Team et al. (2023), or other lightweight VLMs. Since the fusion architecture itself is lightweight, the overall latency in such settings is dominated by the chosen captioning model rather than CapFuse-Net. This separation makes it possible to adapt the system to different real-time constraints by selecting an appropriate captioning backend, while the core fusion model remains efficient and stable.

## 5 CONCLUSION

In this work, we introduced CapFuse-Net, a multimodal framework that integrates pretrained vision-language models with a structured fusion pipeline consisting of the Cross-Feature Fusion Module (CFM), Guided Cross-Attention, and Differential Attention. By augmenting tweet text with LLaVA-generated, image-grounded captions, our method enriches sparse textual inputs and improves cross-modal alignment. Extensive experiments on CrisisMMD and transfer to DMD show that CapFuse-Net consistently outperforms unimodal baselines and strong multimodal systems, achieving state-of-the-art results on informativeness and humanitarian categorization. Ablation studies further demonstrated the central role of CFM, while Guided Cross-Attention and Differential Attention refine multimodal reasoning. Grad-CAM visualizations further demonstrated improved grounding in crisis-relevant regions, enhancing interpretability. Overall, our results demonstrated the importance of carefully designed fusion strategies, coupled with vision-language augmentation for improved multimodal crisis classification across diverse tasks. Future efforts will extend this work to multilingual data and real-time deployment for reliable disaster response and systematically evaluate different captioning models to quantify their effects on caption quality, robustness, and downstream multimodal performance. We believe this direction opens pathways for building responsible, interpretable, and generalizable multimodal systems for high-stakes decision-making.

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

## A    Interpretability Analysis

Interpretability is a critical aspect of multimodal crisis classification systems, especially when deployed in high-stakes environments such as disaster response. We apply Gradient-weighted Class Activation Mapping (Grad-CAM) to visualize the spatial regions in an image that most influence the model's predictions Selvaraju et al. (2017). The resulting heatmaps reveal the image regions that most influenced the model's prediction, allowing us to qualitatively assess whether the model attends to semantically meaningful areas such as damaged buildings, rescue efforts, or debris.

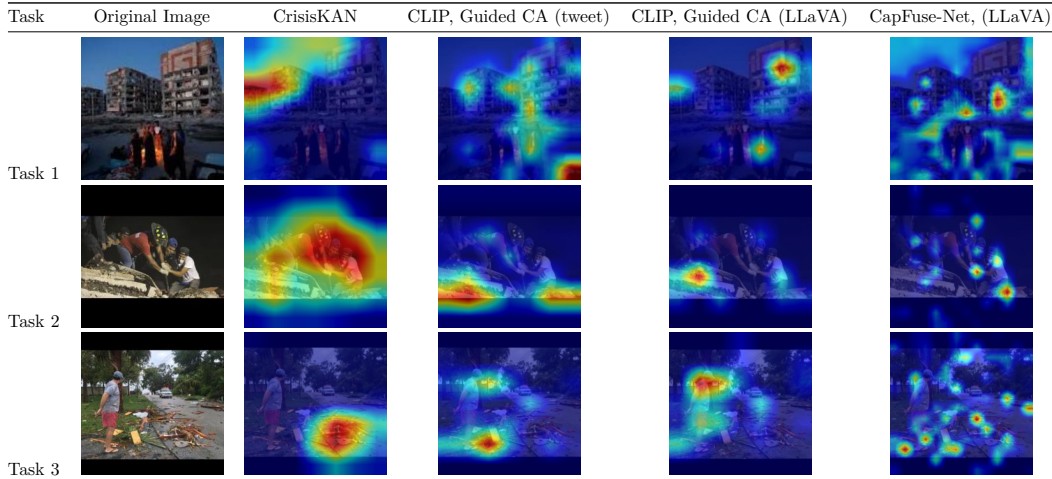

Figure 2: Grad-CAM visualizations of model predictions across the three CrisisMMD tasks. Compared to CrisisKAN, both CLIP with Guided Cross-Attention (using tweet text or LLaVA captions) and CapFuse-Net show improved grounding. In particular, CapFuse-Net (LLaVA) consistently focuses on crisis-relevant regions, demonstrating its ability to attend to informative features critical for accurate classification.

Fig. 2 presents Grad-CAM heatmaps for three representative examples drawn from the crisis classification tasks. We compare visual attention across four model variants: (i)CrisisKAN, (ii) Pretrained CLIP model with Guided Cross-Attention (Guided CA) using tweet text only, (iii) Pretrained CLIP model with Guided CA using tweet text concatenated with LLaVA-generated captions, and (iv) the CapFuse-Net model with the PE visual encoder, CLIP text encoder, and the proposed CFM, incorporating both tweet text and LLaVA captions as input. The CrisisKAN model and the model trained with tweet text (third column) tend to focus on less informative or noisy visual regions. For example, in Task 1, the model incorrectly attends to the corner of the image, ignoring the central scene. In Task 2, although the image depicts a rescue operation, and CrisisKAN focuses on the people, the other model predominantly focuses on the surrounding debris rather than the volunteers. In Task 3, which involves identifying minor structural damage, both models fail to localize any meaningful regions, leading to an incorrect prediction. In contrast, the incorporation of LLaVA-generated captions (fourth column) improves visual grounding by guiding the model's attention to more contextually relevant areas. In Task 1, the model successfully highlights the damaged building; in Task 2, it shows partial attention to the person involved in the rescue; and in Task 3, it attends to the debris scattered on the road, an important visual cue for damage severity. The strongest and most semantically meaningful attention is observed in the CanFuse-Net model (fifth column). This variant consistently attends to critical regions aligned with the target class across all three tasks, such as broken structures, human presence, and localized damage. The focused attention behavior on small details corresponds with the model's superior quantitative performance and suggests that the CFM facilitates improved cross-modal alignment between visual and textual information. These visualizations confirm that our design choices, particularly the

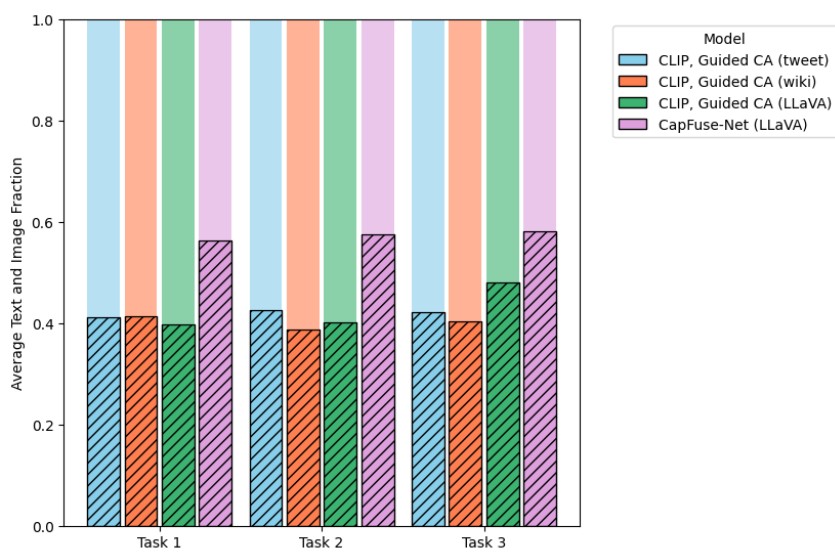

Textured = image fraction, Plain shade = text fraction

Figure 3: Comparison of text and image modality weights across models on the three CrisisMMD tasks. Textured bars represent image fractions, while plain bars represent text fractions, showing how CapFuse-Net (LLaVA) balances contributions from both modalities more effectively.

use of structured text and the CFM, improve both model performance and interpretability by promoting reliance on informative visual evidence.

**Contribution of visual and textual modalities:**   As evident from the unimodal model results, image-only models consistently outperform text-only models across all tasks, suggesting that visual information provides more discriminative features for crisis classification. To further investigate the relative influence of each modality in our multimodal architecture, we conduct a detailed analysis using attribution methods. We utilize the LayerIntegrated-Gradients technique from the Captum library (Kokhlikyan et al., 2020) to compute attribution scores for the input features from both the image and text encoders. This method allows us to estimate the contribution of each modality to the final prediction by attributing relevance scores to the input tokens (for text) and visual embeddings (for image) at the layer level. Specifically, we extract the attribution scores from the final encoding layers of the image and text tokens, just before they are passed into the fusion module. This enables a fair comparison of how much each modality contributes to the model's decision-making process.

Fig. 3 presents the average fractional attribution of each modality across different model variants. Models trained with tweet text or Wikipedia-augmented captions tend to place greater weight on the textual features, while the contribution from visual features is comparatively lower. This imbalance correlates with lower overall performance, suggesting that reliance on textual information limits the model's ability to make accurate predictions, as evidenced by the unimodal models' performances. In contrast, the model trained with our proposed CFM module exhibits a higher reliance on image features. The visual modality contributes significantly to the model's predictions, indicating that CFM successfully enhances visual feature representation and encourages the model to leverage more informative visual cues. This stronger visual attribution aligns with the observed improvement in classification performance, particularly in tasks where visual context is critical, such as identifying damage severity or situational relevance. These findings confirm the importance of effective modality integration and further motivate the need for architectures that can dynamically attend to the most informative modality based on the nature of the input.

(a)                                                    (b)

Figure 4: Baseline unimodal architectures. (a) Image-only Model. (b) Text-only Model

Table 7: Comparison of model performance under standard and class-weighted cross-entropy (W-CE) on the CrisisMMD and DMD datasets.

| Task | Loss | CrisisMMD | | | DMD | | |
|------|------|-----|----------|------|-----|----------|------|
| | | **Acc** | **Macro-F1** | **W-F1** | **Acc** | **Macro-F1** | **W-F1** |
| Task 1 | CE | 93.63 | 92.76 | 93.62 | 89.78 | 89.46 | 89.67 |
| | W-CE | 93.17 | 92.33 | 93.20 | 86.89 | 86.66 | 86.67 |
| Task 2 | CE | 95.72 | 71.54 | 95.30 | – | – | – |
| | W-CE | 95.43 | 83.95 | 95.37 | – | – | – |
| Task 3 | CE | 71.14 | 60.90 | 69.51 | 45.26 | 38.95 | 47.18 |
| | W-CE | 69.45 | 61.47 | 69.78 | 62.90 | 52.60 | 66.62 |

## B    WEIGHTED LOSS ANALYSIS

We also explored a weighted loss function, class-weighted cross-entropy (W-CE), as a strategy to mitigate the class imbalance issue. As reported in Table 7, the effect of weighting varies noticeably across tasks and datasets. For CrisisMMD, W-CE substantially boosts the macro-F1 score in Task 2 (71.54 to 83.95), indicating improved sensitivity to minority humanitarian categories. However, for Task 1, weighting yields a slight drop in both accuracy and F1 metrics. For Task 3, although macro-F1 increases (60.90 to 61.47), accuracy decreases. The pattern is similarly mixed when evaluated on the OOD DMD dataset. W-CE dramatically improves Task 3 performance (for example, accuracy increases from 45.26 to 62.90), but there is a performance drop for Task 1 across all metrics. As the weighted loss did not consistently improve all tasks, we employ the standard cross-entropy objective in our primary experiments to ensure consistency and interpretability of results.

## C    ADDITIONAL RESULTS

In Table 8, we demonstrated the advantage of CapFuse-Net over unimodal and multimodal baselines on the CrisisMMD dataset, reporting classification results in terms of Accuracy, Macro-F1, and Weighted-F1 across all three tasks. Fig. 4 shows the unimodal architectures. We further analyze its predictions through the confusion matrices shown in Fig. 5. These results show that CapFuse-Net achieves strong per-class accuracy, particularly for dominant classes such as other relevant information and severe damage, while performance on minority classes, such as vehicle damage and affected individuals, remains more challenging due to the limited number of samples. We present confusion matrices on the DMD dataset in Fig. 6 after training exclusively on CrisisMMD. For Task 1 (informativeness), CapFuse-Net generalizes well, achieving near-perfect recognition of informative posts, though a portion of non-informative posts are misclassified as informative. For Task 3 (damage severity), the model performs reliably for severe damage and little or no damage, but struggles to separate the intermediate mild damage category, which is often confused with both extremes. These findings demonstrate that CapFuse-Net transfers reliably across datasets, though its per-

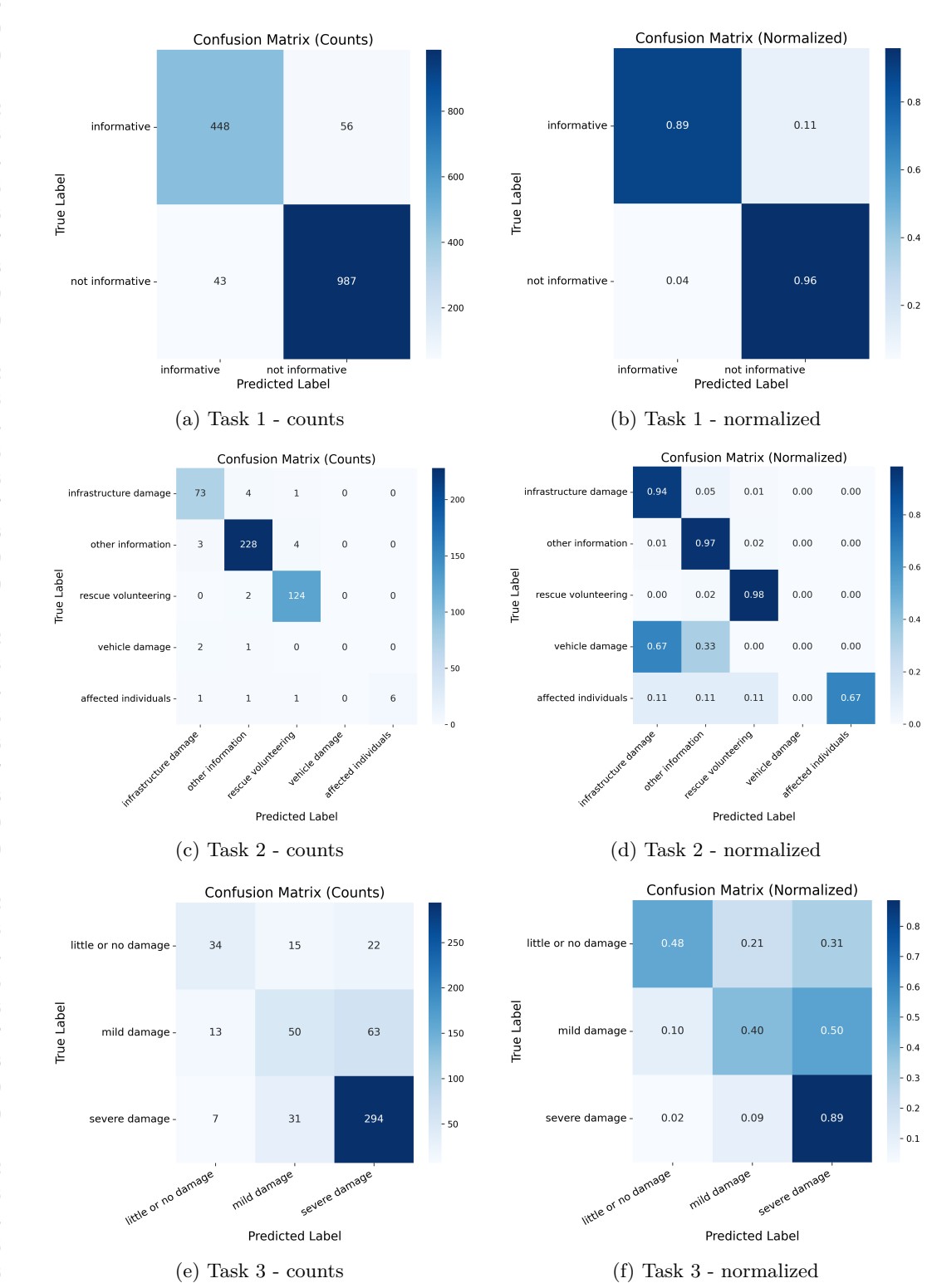

Figure 5: Confusion matrices of CapFuse-Net predictions for the three CrisisMMD classification tasks on the original splits. For each task, the left panel shows raw counts of true vs. predicted labels, while the right panel shows normalized values, highlighting per-class classification accuracy.

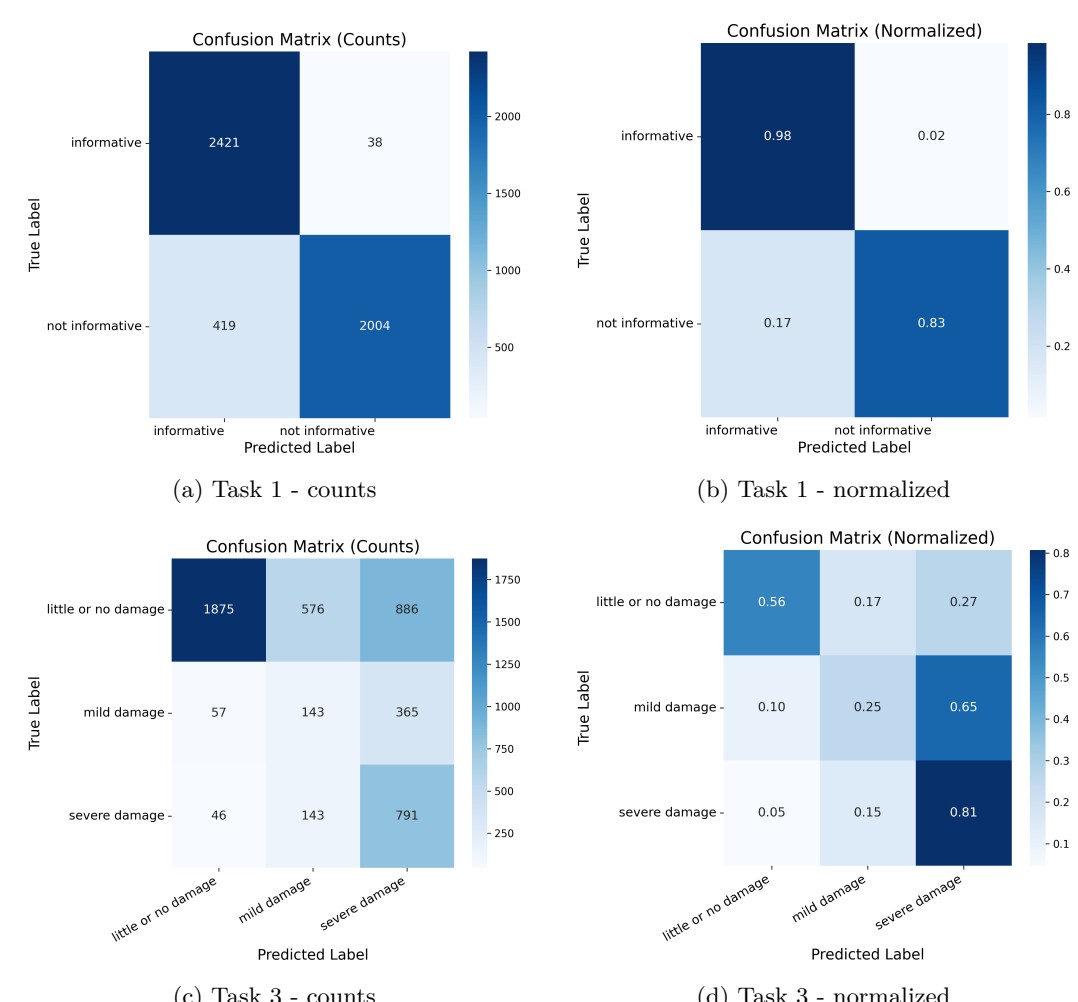

(a) Task 1 - counts

(b) Task 1 - normalized

(c) Task 3 - counts

(d) Task 3 - normalized

Figure 6: Confusion matrices of CapFuse-Net evaluated on the DMD dataset after being trained on CrisisMMD. Subfigures (a) and (b) show counts and normalized results for Task 1 (informativeness classification), where informative posts are recognized with high precision, though some non-informative posts are misclassified as informative. Subfigures (c) and (d) show counts and normalized results for Task 3 (damage severity classification), indicating reliable detection of severe and little/no damage, with mild damage being the most confounded class.

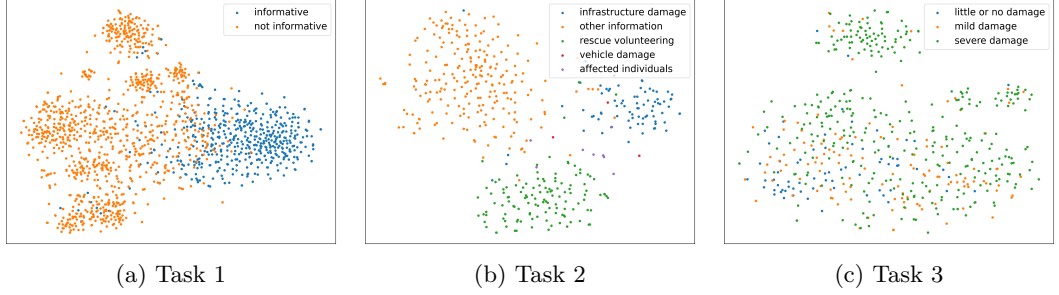

(a) Task 1

(b) Task 2

(c) Task 3

Figure 7: t-SNE visualization of the learned joint embeddings from our proposed CapFuse-Net model across three CrisisMMD tasks. Clear separation in Task 1 and Task 2 indicates that the model produces discriminative and semantically meaningful clusters. Task 3 exhibits a more challenging delineation, especially for the mild or no damage categories.

Table 8: Comparison of unimodal and multimodal models on the three CrisisMMD tasks using different encoders, knowledge fusion methods, and fusion strategies. Results are reported in Accuracy, Macro-F1, and Weighted F1.

| Modality | Vision | Text | KF | Fusion | Task 1 | | | Task 2 | | | Task 3 | | |
|---|---|---|---|---|---|---|---|---|---|---|---|---|---|
| | | | | | Accuracy | Macro F1 | W-F1 | Accuracy | Macro F1 | W-F1 | Accuracy | Macro F1 | W-F1 |
| Image-only | DenseNet | - | - | - | 82.89 | 80.81 | 82.98 | 86.25 | 61.39 | 85.78 | 62.57 | 49.63 | 62.00 |
| Image only | CLIP Vision | - | - | - | 89.20 | 87.77 | 89.20 | 91.43 | 54.66 | 90.22 | 69.50 | 57.52 | 67.86 |
| Image only | PE Vision | - | - | - | 91.07 | 89.86 | 91.06 | 91.28 | 57.32 | 90.26 | 70.70 | 59.64 | 69.33 |
| Text-only | - | Electra | Wiki | - | 84.64 | 81.70 | 84.22 | 87.36 | 60.46 | 87.45 | 62.45 | 50.37 | 62.63 |
| Text only | - | CLIP text | tweet | - | 87.33 | 85.24 | 87.15 | 86.10 | 50.88 | 85.20 | 61.81 | 34.12 | 52.67 |
| Text only | - | CLIP text | Wiki | - | 83.53 | 79.90 | 82.83 | 83.15 | 48.46 | 81.77 | 59.48 | 36.29 | 52.80 |
| Text only | - | CLIP text | LLaVA | - | 87.64 | 85.63 | 87.47 | 86.33 | 50.93 | 85.40 | 62.13 | 34.02 | 53.06 |
| Multi-modal | DenseNet | Electra | Wiki | Guided CA | 86.80 | 85.25 | 86.87 | 91.34 | 66.08 | 91.22 | 64.65 | 44.64 | 61.03 |
| Multi-modal | DenseNet | Electra | Wiki | Cross Attention | 87.32 | 85.71 | 87.36 | 89.28 | 62.53 | 88.82 | 63.07 | 42.97 | 59.32 |
| Multi-modal | DenseNet | Electra | Wiki | Guided CA+Self Attn | 88.36 | 87.00 | 88.44 | 91.43 | 60.25 | 90.75 | 63.83 | 44.95 | 60.92 |
| Multi-modal | DenseNet | Electra | Wiki | Cross Diff Attn | 85.74 | 85.74 | 83.99 | 86.55 | 51.42 | 85.51 | 61.69 | 41.21 | 58.19 |
| Multi-modal | DenseNet | Electra | Wiki | Guided CA+Diff Attn | 89.33 | 87.94 | 89.35 | 91.58 | 57.44 | 90.68 | 63.14 | 46.89 | 61.13 |
| Multi-modal | CLIP Vision | CLIP text | Wiki | Guided CA | 90.57 | 89.05 | 90.45 | 94.02 | 70.95 | 93.65 | 68.94 | 53.50 | 65.69 |
| Multi-modal | CLIP Vision | CLIP text | Wiki | Guided CA+Diff Attn | 90.44 | 89.06 | 90.39 | 93.72 | 71.04 | 93.37 | 68.68 | 53.04 | 65.55 |
| Multi-modal | CLIP Vision | CLIP text | LLaVA | Guided CA+Diff Attn | 92.52 | 91.40 | 92.47 | 93.87 | 68.87 | 93.44 | 68.87 | 55.11 | 66.62 |
| Multi-modal | CLIP Vision | CLIP text | LLaVA | Guided CA | 92.91 | 91.91 | 92.89 | 93.92 | 69.45 | 93.69 | 69.00 | 55.14 | 66.28 |
| Multi-modal | PE Vision | PE text | LLaVA | Guided CA | 91.29 | 90.09 | 91.27 | 94.38 | 68.10 | 93.84 | 70.89 | 58.43 | 68.54 |
| Multi-modal | PE Vision | CLIP text | LLaVA | Guided CA | 92.98 | 92.01 | 92.97 | 94.97 | 68.76 | 94.53 | 70.19 | 57.02 | 67.34 |
| Multi-modal | CLIP Vision | CLIP text | LLaVA | CFM + Guided CA | 92.33 | 91.20 | 92.28 | 94.01 | 69.67 | 93.59 | 70.51 | 58.13 | 67.66 |
| Multi-modal | PE Vision | CLIP text | LLaVA | CFM + Guided CA | 93.63 | 92.76 | 93.62 | 95.72 | 71.54 | 95.30 | 71.14 | 60.90 | 69.51 |

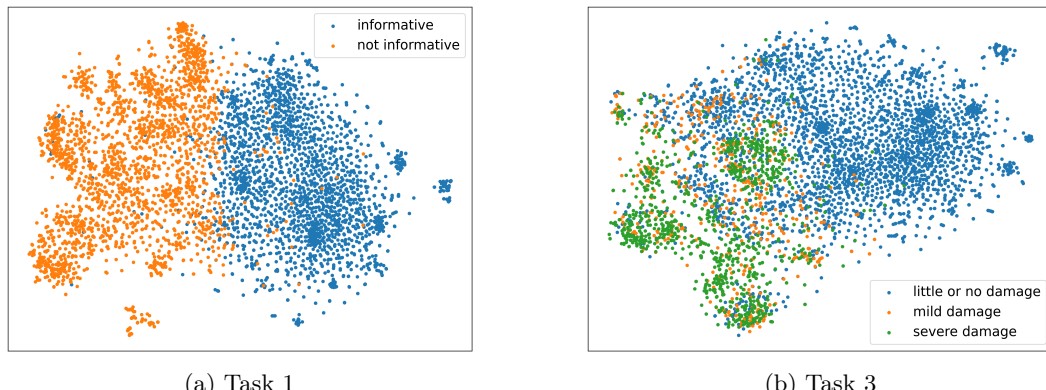

(a) Task 1          (b) Task 3

Figure 8: t-SNE visualization of joint embeddings learned by CapFuse-Net on the DMD dataset, transferred from CrisisMMD. (a) Task 1 shows a clear separation between informative and non-informative posts. (b) Task 3 illustrates partially overlapping clusters for damage severity levels, reflecting both the challenge of the task and the model's ability to capture severity distinctions.

formance is still constrained by data imbalance and the inherent difficulty of distinguishing intermediate damage levels.

Fig. 7 presents the t-SNE visualization of the joint embeddings learned by CapFuse-Net across the three CrisisMMD tasks. For Task 1, the embeddings of informative and non-informative form two clearly separable clusters, indicating that the model effectively captures the binary distinction. In Task 2, categories such as infrastructure damage and rescue volunteering exhibit well-formed clusters, while classes such as other information and affected individuals appear more dispersed and overlap. This pattern illustrates both the inherent semantic ambiguity among certain humanitarian categories and the limited availability of training data, which makes their boundaries harder to distinguish. Task 3 shows less distinct boundaries, particularly between mild and severe damage, which aligns with the subjective and visually subtle nature of severity assessment. To summarize, the visualizations demonstrate that CapFuse-Net produces discriminative and semantically meaningful embeddings, with separability strongest in binary settings and more challenging in fine-grained classification scenarios. We also show the t-SNE visualization of joint embeddings learned by CapFuse-Net on the DMD dataset after being trained on CrisisMMD in Fig.8. For Task 1, the embeddings of informative and non-informative form two largely separable clusters, confirming that the model generalizes well in capturing the binary informativeness distinction even under distribution shift. In contrast, Task 3 exhibits partially overlapping clusters for the three severity levels. We observed a similar pattern in the CrisisMMD t-SNE plots, suggesting that the difficulty in distinguishing mild from severe damage is not

Table 9: Data distribution across the three tasks in the CrisisMMD dataset, showing the number of samples in the training, validation, and test sets, along with the total count for each task.

| Task | Train | Validation | Test | Total |
|---|---|---|---|---|
| Informativeness | 9599 | 1573 | 1534 | 12706 |
| Humanitarian | 2874 | 477 | 451 | 3802 |
| Damage Severity | 2468 | 529 | 529 | 3526 |

specific to dataset bias but reflects the inherent semantic ambiguity of severity assessment. This cross-dataset consistency emphasizes both the robustness of CapFuse-Net in learning transferable embeddings and the persistent challenge of fine-grained damage classification.

## D   DATASET

**CrisisMMD dataset:** The CrisisMMD dataset Alam et al. (2018) is a large collection of Twitter posts from seven major natural disasters in 2017 (hurricanes, earthquakes, floods, and wildfires). Each post contains both textual and visual information and is manually annotated for three classification tasks. Task 1 is the binary classification of distinguishing informative posts from non-informative ones. Task 2 assigns each example to one of five humanitarian categories: infrastructure damage, vehicle damage, rescue efforts, affected individuals (encompassing injuries, fatalities, missing persons, and those found), or others. Task 3 assesses damage severity, labeling instances as severe, mild, or little/no damage. Table 9 summarizes the distribution of training, validation, and test samples across the three CrisisMMD tasks. In Fig. 9, we illustrate the number of samples for each class across the train, validation, and test sets for all three tasks. The figure clearly demonstrates a high degree of class imbalance in the CrisisMMD dataset. For Task 1, the distribution is skewed toward the non-informative class, which significantly outnumbers the informative samples. In Task 2, several classes contain very few instances. To mitigate this extreme sparsity, we grouped three underrepresented categories: affected individuals, injured or dead people, and missing or found people, into a single class during training. Similarly, Task 3 exhibits notable class imbalance, with the severe damage category having substantially more samples than the mild damage and little or no damage classes.

**Damage Multimodal Dataset (DMD):** The Damage Multimodal Dataset (DMD) Mouzannar et al. (2018) is a smaller but complementary multimodal dataset curated for disaster assessment. It consists of 4,882 social media posts containing paired images and text, collected during real-world emergencies such as earthquakes, hurricanes, and wildfires. Each instance is annotated with three labels: informativeness, humanitarian categories, and damage severity. Although the informativeness and severity annotations align with those in CrisisMMD, the humanitarian category schema in DMD follows a different annotation scheme, making direct comparison between datasets difficult. To ensure consistency in cross-dataset experiments, we therefore focus on Task 1 (informativeness classification) and Task 3 (damage severity classification). Although smaller in size compared to CrisisMMD, DMD is more specialized and remains a valuable benchmark for evaluating the transferability and robustness of multimodal crisis models across datasets.

**N24News:** N24News dataset Wang et al. (2021b) contains multimodal news articles with paired images and text, labeled across 24 topical sections (e.g., Sports, Economy, Health). For this experiment, we randomly sampled 10% of the available data for each split, resulting in 4,899 samples for training, 613 for validation, and 613 for testing.

## E   CAPTION AUGMENTATION

We augmented text-captions for all the CrisisMMD data using the LLaVA pretrained model (*microsoft/llava-med-v1.5-mistral-7*) from the HuggingFace model hub. Fig. 10 presents the full system prompt used to guide the LLaVA model in generating detailed and context-

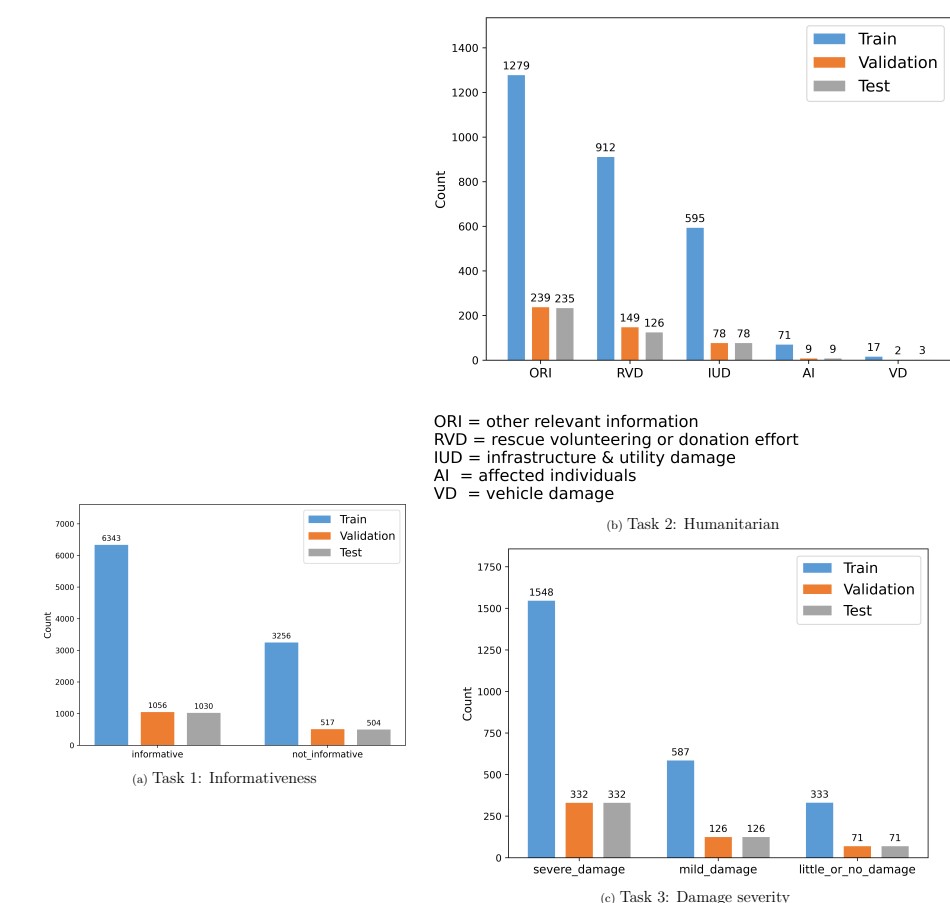

Figure 9: Class-wise train/test/validation splits of the CrisisMMD dataset for all three tasks.

```
<|im_start|>system
- You are LLaVA, a large multimodal assistant trained by UW Madison WAIV Lab.
- You can understand and analyze visual content provided by the user and assist
with a variety of tasks using natural language.
- Follow the instructions carefully and provide detailed, context-aware
explanations.  <|im_end|>
<|im_start|>user
Given the caption:  <tweet_text>, analyze the corresponding image and describe
it in a very detailed and informative manner, focusing on crisis-relevant
visual elements such as damage level, people, infrastructure, or rescue efforts.
<image> <|im_end|> <|im_start|>assistant
```

Figure 10: Instruction prompt used for generating image description using the LLaVA model.

aware descriptions of disaster-related imagery. The prompt is structured using special tokens $(<|im\_start|>, <|im\_end|>)$ to delineate roles and message boundaries for system, user, and assistant interactions.

### E.1    IMPLEMENTATION DETAILS FOR HALLUCINATION SUPPRESSION

For hallucination mitigation, we integrate the method of Jiang et al. (2025) into our captioning pipeline using LLaVA-1.5 (7B) with a ViT-L/14 visual encoder. The approach operates

entirely at inference time, hence requiring no retraining. It detects low-grounded object tokens via the Visual Attention Ratio (VAR) computed from middle-layer attention, and suppresses hallucinations by adjusting attention logits to enforce consistency across heads. We follow the default hyperparameters reported in the paper. All remaining details follow Jiang et al. (2025) and the authors' repository.[1]

---

[1] https://github.com/ZhangqiJiang07/middle_layers_indicating_hallucinations

