# OpenReview forum: "Multimodal vision-language models with guided cross-attention for crisis event understanding"
_ICLR.cc/2026/Conference — Submitted to ICLR 2026_

### Official Review · Reviewer_xHpf · 2025-10-27

**Soundness:** 3
**Presentation:** 3
**Contribution:** 3
**Rating:** 4
**Confidence:** 4

**Summary:**

This paper proposes a novel multimodal framework, CapFuse-Net, for processing social media data in crisis events, integrating both textual and visual information. The core idea of the framework is to generate image-text pair descriptions using an enhanced Vision-Language Model (VLM), and then combine them with a Cross-Feature Fusion Module (CFM), Guided Cross-Attention Mechanism, and Differential Attention Mechanism to improve crisis event classification performance.

**Strengths:**

1. The CapFuse-Net model proposed in this paper cleverly combines image descriptions generated by the Vision-Language Model with textual content.
2. The paper thoroughly validates the effectiveness and robustness of the proposed method through experiments on multiple crisis classification benchmark datasets.
3. Ablation experiments clearly demonstrate the contribution of each module, further supporting the effectiveness of the proposed approach.

**Weaknesses:**

1. CapFuse-Net introduces several complex mechanisms (such as CFM, Guided Cross-Attention, and Differential Attention) in the multimodal fusion process, which may result in a high computational cost. This could become a performance bottleneck, especially when real-time processing of large amounts of social media data is required.

2. Although the model achieves excellent performance on most tasks, it performs poorly in certain tasks (such as damage severity classification) when handling categories with moderate damage. Future work could consider using more diverse crisis event datasets or collecting additional data from actual disaster response scenarios to further validate the model's generalization capability.

3. The paper enhances textual information by generating image descriptions using LLaVA, but the LLaVA model may introduce the risk of generating false information (i.e., "hallucinations"). How to control this risk requires further discussion.

**Questions:**

Please refer to the weaknesses.

---

> ### Author Response · Authors · 2025-11-22
> **Response to Reviewer xHpf**
>
> We thank the reviewer for their thorough review and constructive feedback on our paper.
>
> **W1. “Computational Cost:”**
>
> We thank the reviewer for raising this concern.
> Although CapFuse-Net includes several fusion components (CFM, Guided Cross-Attention, Differential Attention), our empirical measurements show that these modules do not introduce a significant computational bottleneck during inference.
> To quantify this, we measured end-to-end inference latency using a batch size of 16. The results are:
> - Task 1: 0.128 s per batch
> - Task 2: 0.131 s per batch
> - Task 3: 0.123 s per batch
>
> These timings demonstrate that the additional attention layers incur minimal overhead, as inference remains well under 0.14 seconds per batch across all tasks. Combined with the fact that CapFuse-Net adds only ~2.96M trainable parameters (far fewer than CrisisKAN at 129M and MCAModel at 165M), the overall computational footprint of our fusion architecture is lightweight.
>
> We acknowledge that LLaVA captioning is slower and may not be ideal for real-time processing. However, the captioning component is modular and can be replaced with faster image-captioning backends such as GPT-4o or Gemini for deployment scenarios requiring low latency. We will clarify this and include the parameter and inference-time results in the revised version.
>
>
>
>
>
> **W2. “Model’s Generalizability:”**
>
> We thank the reviewer for highlighting this important point. We agree that moderate-damage classification remains a challenging case, not only for CapFuse-Net but also for prior multimodal crisis models. This difficulty stems from the inherent ambiguity of the “moderate damage” category: the visual cues often overlap with both “no damage” and “severe damage,” and the textual descriptions in many cases do not explicitly reference the degree of damage.
> This limitation reflects dataset-level properties rather than model-specific failure modes.
>
> As the reviewer suggests, more diverse or fine-grained crisis datasets could help address this challenge. To this end, we are currently working on collecting new crisis datasets that include: specific disaster-prone geographic regions and event types, and finer-grained annotations for damage severity. These steps directly support the reviewer’s recommendation and will enable a more rigorous evaluation of the model’s generalization in real-world disaster scenarios.
>
>
>
> **W3. “Hallucination Risk:”**
>
> We thank the reviewer for highlighting this concern. We agree that LLaVA-generated captions may introduce hallucinated or spurious content, which can negatively affect downstream multimodal models. To address this, the paper includes an initial hallucination-mitigation step inspired by Jiang et al. (2025), and its impact is evaluated through an ablation study in Table 3.
>
> The ablation compares CapFuse-Net with and without hallucination mitigation under the same fusion configuration. The mitigated captions yield improved performance for Task 3 (damage severity), indicating that filtering hallucinations does provide measurable benefits.
>
> We acknowledge, however, that hallucination control remains an open challenge for VLMs. In the revision, we will expand the discussion to clarify potential strategies such as leveraging faster high-fidelity captioning backends (e.g., GPT-4o, Gemini). We appreciate the reviewer’s suggestion and will include a more explicit discussion of hallucination risks and mitigation strategies in the final manuscript.

---

### Official Review · Reviewer_VHa6 · 2025-10-27

**Soundness:** 3
**Presentation:** 3
**Contribution:** 2
**Rating:** 4
**Confidence:** 4

**Summary:**

In this paper, the authors introduce a novel multimodal framework, named CapFuse-Net, to enhance the understanding and classification of crisis events from multimodal social media data. The main contributions include: 1) a novel multimodal framework that augments sparse textual data with VLM-generated captions for richer context; 2) an innovative Cross-Feature Fusion Module (CFM) that effectively merges original and generated text using cross-attention ; and 3) a guided fusion pipeline combining Guided Cross-Attention and Differential Attention to refine the alignment between visual and textual features while suppressing noise.

The motivation for this work stems from the observation that existing multimodal models often struggle to effectively leverage the complementary nature of visual and textual data in crisis scenarios, particularly when dealing with noisy and information-sparse social media posts. To address this, the authors propose a framework that first enriches the textual input by generating image-grounded captions using a Vision-Language Model. The core innovations of their method include the Cross-Feature Fusion Module (CFM), which fuses the original tweet with the generated caption to create a richer text representation. This is followed by a Guided Cross-Attention mechanism for fine-grained interaction between visual and textual features, and a final Differential Attention layer to enhance salient features while suppressing noise.

Extensive experiments demonstrate that the proposed model, CapFuse-Net, achieves superior performance across multiple crisis classification benchmarks, consistently outperforming existing state-of-the-art baselines. Furthermore, the interpretability analysis, including Grad-CAM visualizations, effectively shows that the model learns to focus on crisis-relevant visual regions, confirming the effectiveness of its attention mechanisms. The key contributions of this work include the CapFuse-Net framework that effectively enriches sparse text and improves cross-modal alignment, the introduction of the Cross-Feature Fusion Module (CFM) for enhanced textual representation, and the design of a multi-stage, attention-guided fusion pipeline that successfully improves both the performance and interpretability of crisis event classification models.

**Strengths:**

1. The authors employ a Vision-Language Model to generate image-grounded captions, effectively enriching sparse social media text. A novel Cross-Feature Fusion Module then integrates this new context to create a superior textual representation for analysis.

2. The model utilizes a sophisticated fusion pipeline with Guided Cross-Attention and Differential Attention for effective cross-modal alignment. This advanced architecture leads to state-of-the-art performance while simultaneously improving model interpretability.

**Weaknesses:**

1.The paper's experiments fail to demonstrate the effectiveness of the Differential Attention module in Table 2, as its inclusion sometimes degrades performance. More tellingly, the authors' final, best-performing model configuration omits this module, indicating it is not an essential component.

2.The paper neglects to analyze the critical trade-off between performance gains and the high computational cost of relying on a large VLM, which raises significant concerns about the model's practical feasibility in real-time crisis response scenarios.

3.The framework's novelty is somewhat diminished as it primarily relies on assembling existing and widely-used modules, such as cross-attention, rather than introducing fundamentally new architectural concepts.

**Questions:**

1.Given that the Differential Attention module did not improve experimental performance and was omitted from the final model, could you clarify its contribution to the overall architecture?


2.Considering the significant computational overhead from the VLM, can you analyze the trade-off between performance gains of the classification task and the feasibility of deploying the model in real-time crisis response scenarios?

---

> ### Author Response · Authors · 2025-11-24
> **Response to Reviewer VHa6**
>
> We thank Reviewer VHa6 for the detailed evaluation and constructive feedback. We appreciate the recognition of our contributions toward enriching sparse crisis text using image-grounded captions and for highlighting the strengths of the Cross Feature Fusion Module and our overall fusion pipeline.
>
>
> **W1. “Effectiveness of the Differential Attention:”**
>
> We thank the reviewer for the thoughtful comment. We introduced the Differential Attention module to examine whether an additional refinement stage could highlight subtle crisis cues such as localized damage patterns, debris, or structural cracks that Guided Cross Attention alone might not fully capture. Crisis images are often visually cluttered, so we considered it important to evaluate whether deeper attention refinement could improve fine-grained reasoning.
>
> As shown in Table 2, Differential Attention improves some tasks but does not provide consistent gains across all settings. Based on these mixed outcomes, we removed it from the final model to preserve simplicity, improve stability, and reduce computational cost. However, including Differential Attention in the ablation study was still valuable. It allowed us to map the design space more completely, identify which components contribute reliably, and highlight opportunities for future extensions. This evaluation also offers a useful foundation for future studies that may focus on more fine-grained damage analysis or richer multimodal reasoning.
>
> **W2. “Computational Cost:”**
>
> We thank the reviewer for highlighting the importance of computational efficiency for real-time crisis response systems. We acknowledge that LLaVA-based caption generation is relatively slow and may not be ideal for strict low latency settings. However, this captioning step is performed only once during offline preprocessing and is not part of the inference pipeline. The goal of this work is to study a fusion architecture that benefits from image-grounded textual enrichment, rather than optimizing the speed of the captioning component itself. CapFuse-Net is deliberately modular: it can accept captions produced by any vision language model.
>
> For real-time deployment, the captioning module can be replaced with faster API based VLMs such as GPT 4o or Gemini, which provide significantly lower latency and high-quality descriptions that scale to streaming data. Once captions are generated, inference relies only on a CLIP-based visual encoder, a lightweight text encoder, and our fusion layers. We measured an end-to-end latency of roughly 0.12 to 0.13 seconds per batch of 16 samples across all tasks, and the model includes only about 2.96 million trainable parameters. These measurements show that the fusion architecture incurs minimal overhead and remains suitable for real-time classification once captions are available. We clarify this modular design and the offline nature of captioning in the revised version.
>
> **W3. “Framework's Novelty:”**
>
> We thank the reviewer for this thoughtful point. While CapFuse-Net builds on established attention mechanisms, its contribution lies in the way these components are combined and adapted to address challenges that are unique to crisis imagery and social media text. Crisis data often contains visually cluttered scenes, incomplete or noisy textual descriptions, and highly variable event contexts. Designing a fusion strategy that can reliably integrate caption-enriched text with visual features under these conditions requires more than just assembling standard modules.
>
> The proposed Cross Feature Fusion Module strengthens weak or ambiguous tweet text by aligning it with VLM-generated captions, producing a richer and more stable textual representation. Guided Cross Attention then uses this enhanced text to focus the visual encoder on disaster-relevant regions, which is essential for detecting small and subtle cues such as debris, cracks, smoke, or partial structural damage. Our ablation study demonstrates that these components work together to provide consistent and meaningful improvements over conventional multimodal approaches. Overall, the novelty of CapFuse-Net comes from its crisis-driven fusion design, which offers a targeted solution to the unique challenges of multimodal crisis analysis.

---

> > ### Author Response · Authors · 2025-11-24
> > **Response to Reviewer VHa6 Questions**
> >
> > **Q1. “Role of the Differential Attention Module:”**
> >
> > We thank the reviewer for this question. The Differential Attention module was included to examine whether an additional refinement step could enhance the fused representations produced by the Cross Feature Fusion Module and Guided Cross Attention. Crisis images often contain cluttered backgrounds and subtle visual cues, so we considered it important to test whether a second stage of attention refinement could amplify small but meaningful features such as localized damage patterns, debris, or partial structural changes.
> >
> > As shown in our ablation study, this refinement helps in a few settings but does not consistently improve performance across all tasks. For this reason, we removed it from the final model to keep the architecture stable and efficient. Even though it is not part of the final configuration, evaluating this module played an important role in shaping the design. It helped us identify which fusion components provide reliable gains and which ones may introduce noise or unnecessary complexity. The insights from this experiment also point to potential future directions, especially for tasks that require finer grained reasoning or more challenging crisis scenarios.
> >
> > **Q2. “Real Time Feasibility:”**
> >
> > We thank the reviewer for raising this question. We agree that a large Vision Language Model introduces computational overhead when generating captions. To clarify, this captioning step is performed only once during offline preprocessing and is not part of the inference pipeline. The goal of our work is to study how enriched, image-grounded text can improve multimodal fusion, rather than optimizing the speed of the captioning module itself. CapFuse-Net is modular by design, and the fusion architecture can accept captions from any image captioning model.
> >
> > For deployment in real-time crisis response systems, the captioning module can be replaced with faster API based VLMs such as GPT 4o or Gemini, which provide low latency and high-quality descriptions for streaming data. Once captions are generated, CapFuse Net’s inference pipeline relies only on a CLIP based visual encoder, a lightweight text encoder, and our fusion components. We measured an end-to-end inference time of approximately 0.12 to 0.13 seconds per batch of 16 samples across tasks, and the model adds only about 2.96 million trainable parameters. These measurements show that the fusion architecture itself is efficient and suitable for time sensitive settings once captions are available. We will make this tradeoff between offline preprocessing and online inference clearer in the final revision.

---

### Official Review · Reviewer_J6NZ · 2025-10-31

**Soundness:** 3
**Presentation:** 2
**Contribution:** 2
**Rating:** 4
**Confidence:** 3

**Summary:**

This paper introduces CapFuse-Net, a multimodal crisis event classification framework that leverages vision–language models (VLMs) within a structured fusion architecture. The approach enriches textual inputs (e.g., social media posts) with VLM-generated image captions, then integrates the original and augmented text using a Cross-Feature Fusion Module (CFM). Multimodal alignment is further improved through Guided Cross-Attention and Differential Attention mechanisms. Experiments on the CrisisMMD, DMD, and N24News benchmarks demonstrate state-of-the-art results across informativeness classification, humanitarian category recognition, and damage severity assessment. Comprehensive ablation studies and interpretability analyses (e.g., Grad-CAM visualizations) confirm the effectiveness and generalizability of each component in the proposed framework.

**Strengths:**

1. Comprehensive Evaluation: Extensive multi-dataset and multi-task benchmarking against strong baselines (e.g., CaMN, CrisisKAN) demonstrates consistent and significant performance gains across settings.
2. Interpretability: Grad-CAM visualizations (Fig. 2) and modality attribution analyses (Fig. 3) offer clear insights into how different modalities contribute to the model’s decisions.
3. Robustness Analysis: Cross-dataset and split-wise evaluations (Table 4) confirm the model’s stability and resilience under distributional shifts.
4. Ablation Rigor: Detailed module-level ablations (Table 3) systematically quantify the contribution of each architectural component to overall performance.

**Weaknesses:**

1. Novelty Gap: The proposed fusion strategy closely parallels existing approaches (e.g., CFM vs. CAM), offering incremental refinements rather than a fundamentally new paradigm.
2. Data Limitations: While CrisisMMD’s class imbalance (Fig. 9) is acknowledged, the paper does not implement or evaluate mitigation strategies such as resampling or loss reweighting.
3. Computational Cost: The framework depends on LLaVA-based captioning and multiple attention modules, yet omits analysis of efficiency metrics (e.g., inference latency) critical for real-time crisis response.
4. Hallucination Mitigation: The integration of Jiang et al. (2025) appears superficial, as no ablation study is provided to assess its necessity or its influence on caption fidelity.

**Questions:**

1. Model Efficiency: How does CapFuse-Net’s computational efficiency compare to simpler fusion paradigms (e.g., early or late fusion)? Do the observed performance gains justify its architectural complexity, particularly for resource-constrained or real-time deployments?
2. VLM Choice: Were alternative vision–language models (e.g., BLIP, GPT-4V) evaluated for caption augmentation? If not, how might the choice of VLM influence the quality and downstream performance of the fused representations?
3. Modality Attribution: The modality attribution analysis (Fig. 3) indicates stronger reliance on visual cues. Does this suggest that the framework’s advantage lies primarily in mitigating textual noise rather than achieving deeper multimodal integration?
4. Class Imbalance: For minority categories, were any data augmentation or synthetic sampling strategies employed to alleviate imbalance and enhance generalization?

---

> ### Author Response · Authors · 2025-11-22
> **Response to Reviewer J6NZ**
>
> We thank Reviewer J6NZ for careful reading and constructive feedback on our paper. We are glad that the reviewer liked our comprehensive evaluation, interpretability, robustness analysis, and rigorous ablations.
>
> **W1. “Novelty Gap:”**
>
> We thank the reviewer for noting the relationship between CFM and CAM. While CFM is inspired by CAM, our Cross-Feature Fusion Module introduces two key conceptual differences tailored specifically to multimodal crisis data.
> - CAM fuses outputs from two independent pretrained text encoders applied to the same text.
> In contrast, CFM fuses two semantically complementary textual views of the same event. This dual-view fusion allows the model to incorporate missing context from the image into the textual stream, which CAM is not designed to handle.
> - In CFM, we assign the caption embedding as the query and the raw tweet embedding as key/value.
> This asymmetry is motivated by the nature of social media data:
>      - Raw tweets are often short, noisy, and incomplete, lacking descriptive detail.
>      - LLaVA captions contain complementary visual information, providing a richer and more structured linguistic representation of the scene.
>
>  Using the caption as the query allows the model to identify which parts of the tweet embedding are semantically relevant to the image-grounded description. This differs from CAM, which applies cross-attention symmetrically to two encoders processing the same text and does not account for differing information roles between the inputs.
>
>
> **W2. “Data Limitations:”**
>
> We thank the reviewer for highlighting the issue of class imbalance. Our main experiments follow the standard CrisisMMD evaluation protocol, consistent with prior work such as CrisisKAN, CaMN, and MCAModel. Nonetheless, following the reviewer’s suggestion, we conducted additional experiments, trained the same CapFuse-Net model using class-weighted cross-entropy (W-CE) across all three tasks.
>
> |        |        |          | CrisisMMD |          |          |          | DMD |          |
> |:------:|------|:---------:|:----------:|:---------:|:---:|:---:|:----------:|:------:|
> | Task   | Loss   | Acc       | Macro-F1  | W-F1      |          | Acc | Macro-F1 | W-F1 |
> | Task 1 | Normal CE | 93.63 | 92.76 | 93.62 |          | 89.78 | 89.46 | 89.67 |
> |        | W-CE      | 93.17 | 92.33 | 93.20 |          | 86.89 | 86.66 | 86.67 |
> | Task 2 | Normal CE | 95.72 | 71.54 | 95.30 |          | – | – | – |
> |        | W-CE      | 95.43 | 83.95 | 95.37 |          | – | – | – |
> | Task 3 | Normal CE | 71.14 | 60.90 | 69.51 |          | 45.26 | 38.95 | 47.18 |
> |        | W-CE      | 69.45 | 61.47 | 69.78 |          | 62.90 | 52.60 | 66.62 |
>
> Thus, class weighting improves some tasks (e.g., CrisisMMD Task 2-3, DMD Task 3) but degrades others (e.g., CrisisMMD Task 1, DMD Task 1). Because the effect is mixed and our goal is to compare fairly with prior work that also uses standard cross-entropy, we keep the unweighted setting as our main configuration and treat W-CE as a complementary analysis. We will include these weighted-loss results for both CrisisMMD and DMD in the appendix and briefly discuss them in the main text.
>
>
> **W3. “Computational Cost:”**
>
> We thank the reviewer for highlighting the importance of computational efficiency for real-time crisis-response systems. We acknowledge that LLaVA-based caption generation is relatively slow and may not be suitable for strict real-time inference when processing streaming crisis data. To clarify, our focus in this work is on the fusion architecture rather than the speed of the captioning component. CapFuse-Net is intentionally modular: the fusion pipeline accepts any image-grounded textual description. We used LLaVA because it is open-source and widely adopted in recent vision-language research, which ensures reproducibility and supports fair comparisons.
>
> For real-time deployment, this captioning step can be replaced by faster API-based VLMs such as GPT-4o or Gemini, both of which provide significantly lower latency, high-quality descriptions, and the ability to scale to streaming data. This allows CapFuse-Net’s design to remain applicable even in time-sensitive scenarios.
>
> **W4. “Hallucination Mitigation:”**
>
> We thank the reviewer for pointing this out. We clarify that an ablation examining the effect of hallucination mitigation is already included in Table 3. This table compares CapFuse-Net using LLaVA captions and using hallucination-mitigated LLaVA captions under the same architecture and fusion settings. The results show that the mitigated captions yield higher performance for Task 3 (damage severity), which is the task where visual details in captions matter most. For Tasks 1 and 2, the differences are smaller, indicating that these tasks are less sensitive to caption hallucinations.

---

> > ### Author Response · Authors · 2025-11-22
> > **Response to Reviewer J6NZ Questions**
> >
> > **Q1: “Model Efficiency:”**
> >
> > We thank the reviewer for this question. CapFuse-Net adds additional fusion components compared to early/late fusion, but parameter analysis shows that these additions are extremely lightweight relative to the backbone encoders.
> > Model    | #Trainable Parameters | #Total parameters
> > |----------|:-------------------------:|:-------------------:|
> > MCAModel (Task 1)  | 165,401,607 | 165,401,607
> > CrisisKAN (Task 1)  | 129,301,179 | 129,301,179
> > CapFuse-Net (Task 1) | 2,958,339 | 513,783,812
> >
> > As shown in the Task 1 comparison, CapFuse-Net requires only 2.96M trainable parameters, whereas CrisisKAN and MCAModel require 129M and 165M trainable parameters, respectively, illustrating that our model achieves better task performance while maintaining a substantially smaller trainable parameter footprint.
> >
> > We also calculated the inference time. Using a batch size of 16, end-to-end inference latency is:
> > - Task 1: 0.128 s per batch
> > - Task 2: 0.131 s per batch
> > - Task 3: 0.123 s per batch
> >
> > These results show that the fusion modules introduce no substantial runtime bottleneck, and inference remains well below 0.14 seconds per batch. We will include the parameter comparison and inference-time results in the revised manuscript.
> >
> > **Q2. “VLM Choice:”**
> >
> > We thank the reviewer for this question. In this work, we used LLaVA for caption augmentation primarily because it is open-source, reproducible, and widely adopted in recent multimodal research, allowing us to run the full pipeline locally and maintain consistency across experiments. Importantly, CapFuse-Net is modular, and the captioning component can be replaced with any image-grounded VLM. Different VLMs may influence the fused representations through: the level of visual details, hallucination behavior, the linguistic structure, and inference speed, which matters for real-time deployments. In the revision, we will add a discussion noting that evaluating alternative VLMs and especially faster captioning APIs such as GPT-4o or Gemini are a promising direction for improving caption fidelity and latency.
> >
> > **Q3. “Modality Attribution:”**
> >
> > We thank the reviewer for this insightful question. Fig. 3 shows that CapFuse-Net places greater attribution weight on visual features. Importantly, this trend is consistent with what we observe in the baseline results: image-only models substantially outperform text-only models across Tasks 1–3. This indicates that visual information carries the dominant discriminative signal in crisis imagery, particularly for damage severity and humanitarian categorization.
> >
> > However, this does not mean that CapFuse-Net’s advantage is limited to filtering textual noise. Crisis tweets are short, noisy, and highly variable, whereas the corresponding images contain rich, unambiguous evidence (e.g., structural damage, debris, flooding). Thus, higher visual attribution is expected and aligns with the empirical finding that image-only > text-only models.
> >
> > Additionally, the fusion ablations in Tables 1 and 2 show that removing the textual stream reduces performance, and CapFuse-Net outperforms image-only baselines. This demonstrates that the model is not merely suppressing noise; it is leveraging complementary textual signals.
> >
> > The modality attribution pattern is consistent with the inherent information distribution in crisis data. CapFuse-Net’s improvements arise from both mitigating textual noise and achieving stronger text–image alignment than simpler fusion methods.
> >
> > **Q4. “Class Imbalance:”**
> >
> > We thank the reviewer for this question. In this work, we focused on evaluating the proposed fusion architecture under the standard CrisisMMD training setting. While we did not introduce additional data augmentation or synthetic oversampling specifically for minority categories, we did examine the effect of alternative sampling strategies through the experiments reported in Table 4.
> > Table 4 includes results under:
> > - Random splits,
> > - Stratified splits (which explicitly balance class proportions across subsets), and
> > - Event-wise splits (which introduce distribution shift).
> >
> > Across random, stratified, and event-wise splits, CapFuse-Net consistently outperforms the baseline models. This clearly shows that its improvements remain stable even when the sampling procedure changes the composition of the train/validation subsets, despite the test sets themselves being inherently imbalanced.
> > We also conducted additional experiments using class-weighted cross-entropy, which improved certain tasks but did not uniformly outperform the unweighted model.

---

> > ### Comment · Reviewer_J6NZ · 2025-11-25
> >
> > Thank you for the thorough and well-organized rebuttal. Your clarifications address most of the concerns I raised, and I appreciate the additional evidence and reasoning you provided. Below I outline my updated thoughts on the key points.
> >
> > 1）The explanation of the design rationale behind CFM is helpful. Positioning the VLM-generated caption as an independent and complementary view—and then using it as the query over tweet semantics—does make the asymmetrical structure clearer. This framing differentiates your approach from conventional CAM-style designs more than I originally understood. With this clarification, I find the novelty claim reasonable.
> >
> > 2）The extra experiment with Class-Weighted CE is informative. The mixed results suggest that naïve re-weighting is not a reliable fix in this multimodal context and may even harm the feature space. Given this, keeping your current setup for fair comparison makes sense. I recommend documenting these findings in the appendix, even if they are negative results—they will likely be valuable for others working with this dataset.
> >
> > 3）The lightweight nature of CapFuse-Net (~3M trainable parameters) is indeed impressive compared to the baselines. Your argument that the fusion component is efficient while the caption generator can be swapped as needed is fair. That said, for clarity, I suggest distinguishing between the latency of the fusion module itself and the end-to-end latency including caption generation.
> >
> > 4）Thank you for pointing me to the ablation results in Table 3—I missed that earlier. The improvement on Task 3, which relies more on fine-grained visual cues, gives solid support for the usefulness of the hallucination pruning step.
> >
> > Overall, the rebuttal is convincing and resolves the main issues I had. The additional evidence and clearer motivation improve my confidence in the contribution of the work. I am updating my score accordingly.

---

### Author Response · Authors · 2025-11-27
**Revised Manuscript Changes Based on Reviewer Feedback**

We thank the reviewers for their detailed and constructive feedback. We have updated the manuscript accordingly and the key updates are summarized below. We believe these changes substantially strengthen the paper. We welcome the opportunity to make further revisions based on additional feedback.

- **Caption Augmentation and VLM Modularity:**
Clarified in Section 3.2 the modularity of our proposed caption augmentation stage, which is performed offline in our experiments, and can be replaced with faster API-based Multimodal LLMs such as GPT-4o or Gemini for real-time deployment scenarios.

- **Computational Efficiency:**
Added a new table (Table 6) and an expanded discussion in Section 4.4 reporting trainable parameters, total parameters, and inference latency to address reviewers’ concerns about model complexity and runtime.

- **Differential Attention Clarification:**
Expanded Section 4.3 to explain the motivation for Differential Attention, summarizing its mixed ablation results.

- **Weighted Loss Experiments:**
Added a “Weighted Loss Analysis” Appendix section (Section B), including a full results table comparing standard CE and class-weighted CE across the IID CrisisMMD and OOD DMD datasets.

- **Hallucination Clarification:**
Added a short concluding remark in Section 4.3, noting that the difference between LLaVA and LLaVA-mitigated captions is small, indicating that caption hallucinations have a limited impact on the evaluated crisis-classification tasks.

- **Conclusion Update:**
Added a new future work direction mentioning evaluation of alternative captioning models (e.g., GPT-4o, Gemini) and their potential influence on caption quality and downstream performance.

---

### Meta-Review · Area_Chair_EuvT · 2026-01-13

**Summary:**

**Summary**

The paper presents CapFuse-Net, a multimodal framework designed for crisis event classification by integrating visual and textual data from social media. Key contributions include the use of Vision-Language Models (VLMs) to generate image captions that enrich sparse textual inputs, the development of a Cross-Feature Fusion Module (CFM) for effective text merging, and a guided fusion pipeline that enhances multimodal alignment through attention mechanisms. Extensive experiments show that CapFuse-Net outperforms existing models on various benchmarks, and interpretability analyses confirm its focus on relevant visual features, highlighting the framework's effectiveness and generalizability.

**Benefits**
- **Comprehensive Evaluation**: The paper showcases extensive benchmarking across multiple datasets and tasks, demonstrating significant performance improvements over strong baselines like CaMN and CrisisKAN.

- **Interpretability**: The use of Grad-CAM visualizations and modality attribution analyses provides clear insights into the model's decision-making process, enhancing understanding of how different modalities contribute.

- **Robustness Analysis**: The model's stability and resilience are confirmed through cross-dataset and split-wise evaluations, ensuring its effectiveness under varying conditions.

**Weaknesses**
- **Novelty Gap**: The proposed fusion strategy is similar to existing methods, offering only incremental improvements rather than introducing a fundamentally new approach.

- **Data Limitations**: The paper acknowledges class imbalance but fails to implement or evaluate strategies to mitigate this issue, such as resampling or loss reweighting.

- **Computational Cost**: The framework relies on complex components like LLaVA-based captioning and multiple attention modules without analyzing efficiency metrics, raising concerns about its practicality for real-time applications.

- **Hallucination Mitigation**: The integration of existing methods for hallucination mitigation is superficial, lacking an ablation study to evaluate its necessity or impact on caption accuracy.

**Decision**
My recommendation is to review the paper, as several concerns have been raised. Additionally, incorporating more alternative vision-language models would enhance its quality. An ablation study could effectively showcase the advantages of the approach, depending on the performance of the models used.

**Reviewer Concerns:**

All the concerns were addressed and justified, but an additional review process is required before accepting the paper.

**Reviewer Scores:**

Reviewer J6NZ is likely to improve their score, and it is probable that other reviewers will do the same, as many of their concerns have been addressed effectively.

---

### Decision · Program_Chairs · 2026-01-26

Reject